# Local government debt and corporate tax burden: A perspective based on the trade-off of government tax collection and management

**Wei Tang[1], Xingzhu Zhao [2]\*, Shengbao Zhai[3]\*, Lei Cao[1]\***

**1** Auhui University of Finance and Economics, Bengbu, Anhui, China, **2** Dongbei University of Finance and Economics, Dalian, Liaoning, China, **3** Huaibei Normal University, Huaibei, Anhui, China

\* zhaoxingzhu@126.com (XZ); financing@139.com (SZ); accaolei@126.com (LC)

**Data Availability Statement:** Publicly available datasets were analyzed in this study. The financial data is available from the CSMAR database (https://

## Abstract

As an important source of local fiscal revenue, will enterprise tax be affected by local government debt? What role do the government's tax collection and management motives and behaviors play in this effect? By investigating the impact of local government debt on the actual tax burden of enterprises, this study shows that local governments have a trade-off of tax collection in the process of resolving the debt repayment pressure. The study finds that, in general, the expansion of local government debt has increased the actual tax burden of enterprises, which is mainly reflected in non-state-owned enterprises and enterprises that are collected and managed by the local tax department. The results of the mechanism test show that local debt pressure will encourage local governments to adjust the intensity of tax collection and tax incentives, and then increase the level of tax burden of enterprises in the jurisdiction. Furthermore, the heterogeneity test that distinguishes the institutional environment shows that there are significant differences in the taxation behavior of local governments and the impact of the corporate tax burden in different regions. Specifically, the strict tax behavior of local governments is more significant in regions with better institutional environment, while regions with worse institutional environment, due to the lack of market competitiveness, are more inclined to provide a relaxed tax collection environment to enterprises in their jurisdiction, so as to stabilize the tax base and resolve debts through long-term tax growth. In the context of unbalanced regional development, this study provides empirical evidence that the expansion of local debt affects the taxation behavior of local governments, and then affects the actual tax burden of enterprises in the jurisdiction, which is helpful to understand the government behavior during the transition period of developing countries, and provides policy implications for improving the public debt management system, creating a fair tax environment, and promoting high-quality economic growth.

www.gtarsc.com/); data related to local
government debt is from China Government Debt
Center (http://www.celma.org.cn/); other macro-
economic data comes from the National Bureau of
Statistics of China (https://data.stats.gov.cn/).

**Funding:** This work was supported by the National
Social Science Foundation of China under Grant
number 20BJY023. The funders had no role in
study design, data collection and analysis, decision
to publish, or preparation of the manuscript.

**Competing interests:** The authors have declared
that no competing interests exist.

## Introduction

In recent years, local government debt has been rising rapidly and has become an important source of systemic financial risk in China. After the outbreak of the global financial crisis in 2008, China implemented a proactive fiscal policy—a 4 trillion fiscal stimulus plan—to hedge the adverse impact of the crisis. With the implementation of the plan, local governments quickly raised huge amounts of funds through financing platforms to support the quadrillion investment plan, and the scale of local debt began to swell rapidly. Especially after experiencing the European debt crisis in 2012, the trade friction between China and the United States in 2018, and the COVID-19 in 2020, the intertwining and overlapping of unstable internal and external factors make local governments increasingly dependent on raising debt for development. As of the end of December 2020, the national local government debt balance was 25.66 trillion yuan, accounting for about 25.26% of GDP. If the implicit debts are included, the risk of local government debt will be even higher. Since 2018, provincial local government financing vehicles in Yunnan Province, Tianjin Province, and other regions have successively defaulted, making the central government highly concerned about the risk of local government debt repayment.

The local government's debt repayment funds mainly come from land-transferring fees and tax revenue. The expectation of the central government's assistance and its high reliance on land finance are important reasons why local governments dare to borrow excessively [1, 2]. However, these foundations and supports have changed significantly in recent years. On the one hand, since 2014, local government debts have been required to try out the "self-issue and self-repayment" model, and the central government has implemented the "no assistance" principle. On the other hand, the central government has strictly controlled housing prices, making local land fiscal revenue limited. So, in the context of the sharp increase in debt service pressure, have local governments increased the tax burden of enterprises within their jurisdictions? At the same time, the two main taxes (value-added tax (VAT) and enterprise income tax (EIT)) paid by Chinese enterprises have both implemented tax reforms focusing on tax reduction, but the "pain" of Chinese enterprises' tax burden has increased rather than decreased [3]. Is this also related to the expansion of local government debt? From the existing research, this issue has not attracted the attention it deserves. Previous studies on local government debt mostly focused on its causes and macroeconomic consequences [4–6]. With the rapid expansion of the debt scale, the impact of local government debt on micro-enterprises has gradually attracted the attention of scholars [7–9], but few studies have investigated the micro-level consequences of local government debt from the perspective of the corporate tax burden. Yang and Song [10] find that the higher the local government debt, the heavier the tax burden of enterprises through the sample data from 2012 to 2013. However, there is no empirical test on its mechanism, nor does it consider the impact of factors such as the level of economic development and institutional environment differences between regions on the tax collection behavior of local governments.

In fact, on the one hand, although China's nominal tax rate and tax-related laws and regulations are uniformly regulated by the central government, the specific tax collection, administration, and enforcement of tax laws are delegated to grass-roots tax authorities. Therefore, local governments often have discretionary power in tax collection and management. The fiscal decentralization system and the reform of the tax distribution system enable local governments to have the motivation and ability to adjust the intensity of tax collection and management, to achieve "local financing" and participate in "tax competition (a race to the bottom)" [11–13]. Sometimes local finance departments even illegally provide tax rebates for tax competition [14]. On the other hand, in the context of regional competition, local

governments mainly compete for mobile enterprises through tax-collection tools, and tax incentives are often used by local governments as an important strategy for attracting investment [15, 16]. This phenomenon is also fully reflected in China, which is still in the stage of emerging development. Although the tax laws and tax systems in different regions of China are similar, there are great differences in the enforcement strength and discretion among regions, which provides flexibility for local governments to implement tax collection and management according to their actual needs [17]. When faced with fiscal sustainability risks, local governments can choose to strengthen tax enforcement to increase tax revenue, or they can choose to reduce tax collection effort to attract the liquid tax base to expand their fiscal capacity [18–20]. Compared with economically developed areas, due to the lack of competitiveness, local governments in undeveloped areas often need to give more tax incentives to enterprises to retain them or attract other investments [21]. Therefore, to investigate the impact of local government debt on the actual tax burden of enterprises, it is necessary to comprehensively consider the heterogeneity of local government tax enforcement intensity in combination with regional economic development level, institutional environment differences, and other factors.

Based on the above, starting from the motivation and behavior of local governments to intervene in tax collection, this study investigates the impact of local government debt on the actual tax burden of enterprises and finds that there are taxation trade-offs for local governments in resolving debt repayment pressures. The results of the study show that, in general, the expansion of local government debt has increased the actual tax burden of enterprises, which is mainly reflected in non-state-owned enterprises and enterprises that are collected and managed by the local departments (Local Taxation Bureau). Further research shows that local governments affect the tax burden level of enterprises in their jurisdiction by adjusting the intensity of tax collection and tax incentives, and this effect is more significant in areas with a better institutional environment. This result indicates that when local governments faced debt servicing pressure, local governments would make a trade-off between "short-term tax revenue" and "long-term tax revenue". Due to the lack of market competitiveness in areas with poor institutional environments, such areas tend to provide a more relaxed tax collection environment to stabilize the tax base and resolve debt problems through long-term sustainable growth of tax revenue. Therefore, in such areas, the impact of local government debt on the actual tax burden of enterprises is relatively weak.

The possible incremental contributions of this study are reflected in the following two aspects: First, under the realistic background of Chinese-style decentralization and regional competition, based on different enterprise characteristics and the imbalanced regional institutional environment differences in developing countries, this study provides new empirical evidence for the economic consequences of local government debt from the perspective of the micro-corporate tax burden. It enriches the research on regional competition, local government borrowing and local tax collections, and also provides theoretical reference and decision support for preventing and resolving major financial risks. Second, it incorporates government intervention into the research framework, analyzes the motives, conditions, and ways of local government intervention in tax collection, and proposes that there are different situations in which local government debt affects the actual tax burden of enterprises. By further examining the impact of different institutional environments, it shows that local governments have strategic behaviors in tax collection in the process of resolving debts. The above is helpful to understand the government behavior during periods of economic transition in developing countries, and also reveals the role of the institutional environment in regional competition.

The structure of the subsequent chapters is arranged as follows: the second part is the theoretical analysis and hypothesis; the third part is the data source and research design; the fourth

part reports the empirical analysis results and robustness tests; the fifth part reports further analysis results; and the last part is the main conclusions and policy implications.

## Theoretical analysis and hypothesis development

### Local government debt and enterprise tax burden

In 1994, China carried out the reform of the tax distribution system, which weakened the financial power of local governments. But the responsibilities of local governments for infrastructure construction and public services have not been reduced accordingly. In addition, China has long had a promotion and incentive system for officials based on GDP performance assessment. All these have led local governments to issue bonds to develop urban construction and the local economy [22]. The dislocation of financial power, administrative power and expenditure responsibility, as well as the implementation of the 4 trillion fiscal stimulus plan, have led to the rapid growth of the scale of local government debt. Local governments are facing increasing debt repayment pressure and the financial risks and economic security problems caused by debt. "Controlling the increment and transforming the stock" has become the focus of China's policy to resolve government debt risks in recent years.

Some scholars have pointed out that China's local government debt is heavily reliant on land finance, with land transfer revenue being the main source of local government debt repayment [23, 24]. However, due to the exhaustion of land resources in some areas, the increase in land transfer costs, and the strict control of housing prices by the central government, local land fiscal revenue has been greatly restricted [25]. Additionally, China has recently imposed clear restrictions on the use of land transfer income. For example, the repayment liability of Public-Private-Partnership (PPP) projects cannot be linked to the land transfer income. By the end of the "14th Five-Year Plan" period, the proportion of land transfer income used in agriculture and rural areas will be increased to more than 50%. As a result, the heavy responsibility of local government debt repayment will eventually be borne by fiscal revenue sources other than land transfer revenue.

Tax revenue is the largest component of China's fiscal revenue [26]. A number of studies have pointed out that due to the lack of the right to set tax categories and tax rates [18, 19], when local governments face increasing financial pressure and need to increase fiscal revenue, they will increase their own financial resources by adjusting the intensity of regional tax collection [20]. However, the relevant literature has not reached a consensus on the direction of adjustment of tax collection intensity. Some studies believe that when local financial resources are limited, local governments will strengthen tax enforcement to offset short-term negative fiscal shocks [18–20]. Some other studies believe that the decrease of fiscal revenue and the mismatch of fiscal revenue and expenditure will induce tax competition among local governments, which means local governments will ensure the steady growth of fiscal revenue by reducing the intensity of tax collection and attracting the injection of liquid tax base [27, 28]. It can be seen that when facing increasing financial pressure, local governments may choose to strengthen the tax collection intensity and increase fiscal revenue by increasing the actual tax burden of enterprises within their jurisdiction, or they may choose to reduce the tax collection intensity and attract fluid tax base to ensure the continued growth of fiscal revenue. Given the above, this study believes that when local governments face the debt repayment pressure caused by debt expansion, their intervention in tax collection will also face two choices: "short-term" or "long-term".

On the one hand, the government has the motivation to increase fiscal revenue in a short time by strengthening tax collection and management, which increases the actual tax burden of enterprises, due to the pressure of debt repayment and local rigid expenditure. Although

China clearly stipulates that the tax types and tax rates shall be determined by the central government, the tax distribution system, the fiscal decentralization system and the dual leadership and management structure of the tax system (before the merger of the State Taxation Bureau and Local Taxation Bureau) provide sufficient motives and conditions for local governments to intervene in tax collection [29]. In addition, local governments also have certain discretionary powers in tax incentives. For example, preferential tax policies related to regions and industries need to be approved and recognized by government departments. Additionally, there is a phenomenon of tax rebates in the name of industrial support [26, 30]. Zhao et al. [31] point out that the local government's fiscal capacity is a realistic factor restricting regional tax effort, and the financial pressure caused by expenditure demand will make the government strengthen tax collection effort. The fiscal pressure brought by the expansion of the local government debt scale and the growth of expenditure demand caused by debt repayment may cause local governments to intervene in tax collection and tax incentives to alleviate the debt repayment pressure. Local governments may assign higher tax tasks to the tax authorities, and increase fiscal revenue to meet debt repayment needs [13, 32]. These behaviors may increase the actual tax burden of enterprises within their jurisdiction. In light of the above discussion, this study proposes the first hypothesis:

H1(a). The expansion of local government debt will increase the level of the enterprise tax burden.

However, on the other hand, a heavy tax burden could lead to a loss of tax sources in the future, which is not conducive to the establishment of a sound investment environment and could damage the long-term economic development of the region. According to the theory of "voting with feet" [12], enterprises seek a combination of regional public goods and tax policies that meet their preferences [33], and they will grow through continuous migration between regions [34]. The tendency of enterprises to migrate to "Local Tax Heaven" has led local governments to conduct strategic tax collection and management to attract liquid capital. By controlling tax effort, granting tax incentives, or even delaying collections in violation of regulations and providing tax rebates beyond their authority [14, 26, 35], local governments reduce the actual tax burden of enterprises, thereby attracting investment, expanding the tax base, and ensuring the sustainable growth of regional fiscal revenue [31, 36, 37]. Even under intense fiscal pressure, in the context of regional competition, governments may still choose to engage in a race for scarce resources [38]. By reducing the intensity of tax collection and the actual tax burden on enterprises, they can enhance the attractiveness of the regional economy, attract more mobile capital to expand the tax base, and increase the sustainable fiscal revenue in the future. Additionally, the assessment of the level of regional investment attraction also encourages local government officials to "race to the bottom" for future promotion opportunities [39]. Therefore, when local governments face fiscal pressure due to debt expansion, for the consideration of competing for tax base and maintaining sustainable competitiveness, they may choose to provide a relaxed tax environment to participate in tax competition and relieve debt pressure by obtaining long-term fiscal revenue growth. This kind of intervention will reduce the effective tax burden of enterprises in the jurisdiction. Based on the above, this study proposes the second hypothesis:

H1(b). The expansion of local government debt will reduce the level of the enterprise tax burden.

## Impact of different property rights

Under China's unique institutional background, the relationship between local government debt and enterprise tax burden may be influenced by the characteristics of property rights.

First, there is a natural "blood relationship" between state-owned enterprises and the government, and local governments often impose many policy burdens and social responsibilities on these enterprises [40]. Moreover, the executives of state-owned enterprises are often directly appointed by the competent government departments, which gives them certain administrative attributes. This "quasi-official" attribute makes state-owned enterprises more likely to comply with government goals, resulting in weaker tax avoidance motives and relatively high tax burdens [41, 42]. Therefore, when local government increases the tax collection efforts to alleviate debt repayment pressure, state-owned enterprises have less room for the tax increase [10], and their tax burden is less sensitive to local government debt repayment pressure. In contrast, non-state-owned enterprises have stronger tax avoidance motives driven by profit maximization, and often become the focus of tax audits. In addition, due to the lack of inherent political advantages, non-state-owned enterprises are relatively estranged from the government and have weak lobbying ability in the face of tax collection and management [26]. Therefore, if local governments choose to strengthen the tax collection intensity due to the debt expansion, non-state-owned enterprises are more likely to become the target of its sharing pressure.

Second, state-owned enterprises have relatively consistent political goals with local government, and since local governments control the assessment and appointment of state-owned enterprise executives [40], they have stronger control over state-owned enterprise investment behavior. Therefore, the motivation of state-owned enterprises to relocate to other regions is weak, even though their tax collection and management environment is strict. Xu et al. [43] argue that state-owned enterprises are owned by local governments and institutions, making it difficult for them to move to other regions, while private enterprises are not constrained in this way. Therefore, when local governments consider attracting liquid capital and ensuring sustainable growth of fiscal revenue through a "race to the bottom", non-state-owned enterprises are more likely to become the target of their tax base training.

Based on the above, this study proposes the following hypothesis:

H2. The impact of local government debt on enterprise tax burden is more significant in non-state-owned enterprises.

## Impact of different tax authorities

Since the reform of the tax distribution system in 1994, China has implemented a parallel system of the State Taxation Bureau (STB) and the Local Taxation Bureau (LTB) for a long time. The STB is directly under the State Administration of Taxation (which is under the State Council of the People's Republic of China), while the LTB is under the dual leadership of the local government and the State Administration of Taxation, and the primary administrative power is assigned to the local government. Therefore, the strength of tax collection and administration of the STB and the LTB reflects the will of different leading departments and there are significant differences. To be specific, although some departments of STB in some regions collaborate with local financial departments and depend on local governments for basic office conditions [14], the STB is directly subordinate to the State Administration of Taxation and is subject to limited intervention by local governments, while the organizational structure, staffing and management system of the LTB are all managed by the local government, as a result, they are more vulnerable to local government interference [44].

A number of studies have proved that local government intervention has different effects on the tax collection intensity of the STB and the LTB [44, 45]. For example, in 2002, China enacted an income tax-sharing reform whereby the corporate income tax, which was originally levied and administered by the LTB became a central-local shared tax. At the same time, the

reform allocated the power of collecting and managing income tax of newly established enterprises to the STB. This reform had greatly narrowed the scope of tax collection and management of the LTB, weakened the financial power of local governments, and then made local governments have to take remedial actions. Taking this reform as a quasi-natural experiment, Tian and Fan [29] found that the reduction of the corporate income tax sharing ratio would sour local governments to intervene in tax collection and administration. Local governments will reduce the tax effort of tax departments to achieve tax competition. However, this effect only occurred in the enterprises managed by the LTB and not in those managed by the STB. The research confirms that the LTB has become a tool for local governments to regulate the intensity of tax collection and management and realize tax competition. At the same time, it also indirectly proves that the intervention ability of local governments to different tax departments varies. Compared with the vertical management of the STB, because the LTB system implements a dual leadership system, the local government has more voice in it. Therefore, the intervention of local governments will have a greater impact on the enterprises that are levied and managed by the LTB.

Based on the above, this paper proposes the following hypothesis:

H3. The impact of local government debt on enterprise tax burden is more significant in companies levied by LTB.

## Research design

### Sample and data sources

In 2014, China promulgated the new Budget Law and the Regulations of the State Council on Strengthening the Management of Local Government Debt, redefining the scope of local government debt and standardizing the financing model of local governments. China Government Debt Center has released information on the balance of provincial and municipal debt since 2015. Considering the impact of COVID-19 in 2020, Chinese local governments issued a large number of special bonds for epidemic prevention and control (the new local debt in the first quarter of 2020 increased by 35.94% year-on-year). To avoid the impact of the sharp increase in local debt caused by emergencies on the empirical results, this study defines the sample interval as 2015–2019. Select Shanghai and Shenzhen A-share non-financial listed companies, and draw on existing research to process the data as follows: (1) we exclude ST companies, (2) we exclude sample observations with negative owner's equity, negative pre-tax profit, and negative corporate tax expense and those in which the effective tax rate is lower than 0 or greater than 1 [45], (3)we exclude samples with missing data required for the study. The final sample contains 12,325 firm-year observations. The financial data of companies comes from Accounting Research Database (WIND and CSMAR), and macro-economic data is collected from the National Bureau of Statistics of China.

### Regression model

We construct Eq (1) to test our hypotheses:

$$\text{Taxburden}_{i,t} = \alpha_0 + \beta_0 \times \text{GovDebt}_{j,t} + \gamma_0 \times \sum \text{Control}_{i,t} + \varepsilon_0 \qquad (1)$$

where $\text{Taxburden}_{i,t}$ represents the actual tax burden of firm *i* in the year *t*. On the one hand, due to the complexity of corporate financial data confirmation, local governments have more flexibility in the actual operation of corporate income tax collection and management. Facing the fiscal pressure brought by local government debt, corporate income tax may become the first choice for local governments to adjust tax revenue [28, 30, 45]. On the other hand,

turnover tax plays an important role in China's tax system. And Liu and Liu [26] put forward that the tax rebate given by the government is the main channel for the government to manipulate tax management, and it is also the main reason for the difference in corporate tax burden. It shows that the local government also has some discretionary space in the collection and management of other taxes except income tax. Based on these, we draw on existing research to construct two actual tax burden indicators, which represent the actual tax burden of corporate income tax and the comprehensive actual tax burden of enterprises [26, 46]. *Taxburden1* = income tax expense / pre-tax accounting profit, *Taxburden2* = (various taxes paid—tax refund received) / total operating income.

GovDebt$_{j,t}$ indicates the scale of local government debt, expressed by *GovDebt1* (the balance of local debt divided by the GDP of each region). In the robustness test section, it is expressed by *GovDebt2* (the balance of local debt divided by the fiscal revenue of each region).

*Control$_{i,t}$* indicates the control variables in this study, including variables for corporate financial characteristics, regional level and year- and industry-fixed effects. The detailed definitions of the variables are presented in Table 1. In addition, considering our number of clusters ($\approx$30) is relatively small, we use the wild bootstrap method proposed by Cameron et al. [47], clustering the standard errors at the provincial and municipal levels.

## Empirical results

### Descriptive statistics

Table 2 reports the descriptive statistics of the main variables of the article. The statistical results show that, whether it is the actual tax burden of corporate income tax (*Taxburden1*) or the comprehensive actual tax burden of enterprises (*Taxburden2*), there are significant differences in the actual tax burden of enterprises between the samples. The minimum value of

**Table 1. Variable definitions.**

| Variables | Definition |
|---|---|
| *Taxburden1* | income tax expense / pre-tax accounting profit |
| *Taxburden2* | (various taxes paid—tax refund received) / total operating income |
| *GovDebt1* | The balance of local debt divided by the GDP of each region |
| *GovDebt2* | The balance of local debt divided by the fiscal revenue of each region |
| *Namerate* | Nominal tax rate of the firm |
| *Size* | The natural logarithm of a firm's total assets at the fiscal year-end |
| *Roa* | The ratio of return to total assets at the fiscal year-end |
| *BM* | Total assets at the fiscal year end divided by the total market value |
| *Lev* | Total liabilities divided by total assets at the fiscal year-end |
| *Tangible* | The net value of fixed assets divided by total assets at the fiscal year-end |
| *Intangible* | The net value of intangible assets divided by total assets at the fiscal year-end |
| *Invent* | Net value of inventory divided by total assets at the fiscal year-end |
| *Loss* | When the net profit of the company in the previous year is less than 0, the Loss is taken as 1, otherwise, it is 0 |
| *Top10* | The shareholding ratio of top ten shareholders |
| *Indep* | Number of independent directors divided by the total number of directors |
| *Density* | Area population divided by area |
| *Inv* | Social investment in fixed assets divided by regional GDP |
| *GDPr* | Regional GDP growth rate |
| *Year* | Year-fixed effect |
| *Industry* | Industry-fixed effect |

**Table 2. Descriptive statistics.**

| Variables | Observations | Mean | Median | STD | Min | Max |
|---|---|---|---|---|---|---|
| Taxburden1 | 12325 | 0.1864 | 0.1574 | 0.1116 | 0.0071 | 0.6650 |
| Taxburden2 | 9390 | 0.0620 | 0.0507 | 0.0472 | 0.0023 | 0.2513 |
| GovDebt1 | 12325 | 0.1807 | 0.1566 | 0.0753 | 0.0996 | 0.5111 |
| GovDebt2 | 12325 | 1.6415 | 1.5826 | 0.8382 | 0.7002 | 4.6249 |
| Namerate | 12325 | 0.1857 | 0.1500 | 0.0504 | 0.0900 | 0.2500 |
| Size | 12325 | 22.2946 | 22.1108 | 1.3145 | 20.0768 | 26.3522 |
| Roa | 12325 | 0.0531 | 0.0449 | 0.0395 | 0.0020 | 0.1940 |
| BM | 12325 | 0.5255 | 0.4809 | 0.2786 | 0.0867 | 1.1826 |
| Lev | 12325 | 0.3999 | 0.3874 | 0.1938 | 0.0593 | 0.8534 |
| Tangible | 12325 | 0.2030 | 0.1699 | 0.1551 | 0.0024 | 0.6894 |
| Intangible | 12325 | 0.0456 | 0.0338 | 0.0494 | 0.0001 | 0.3260 |
| Invent | 12325 | 0.1374 | 0.1071 | 0.1256 | 0.0003 | 0.6638 |
| Loss | 12325 | 0.0468 | 0.0000 | 0.2113 | 0.0000 | 1.0000 |
| Top10 | 12325 | 0.6078 | 0.6219 | 0.1450 | 0.2636 | 0.9054 |
| Indep | 12325 | 0.3732 | 0.3333 | 0.0518 | 0.3077 | 0.5714 |
| Density | 12325 | 0.0839 | 0.0603 | 0.0942 | 0.0014 | 0.3848 |
| Inv | 12325 | 0.0902 | 0.0898 | 0.0225 | 0.0401 | 0.1521 |
| GDPr | 12325 | 0.0869 | 0.0871 | 0.0518 | -0.1922 | 0.2022 |

*GovDebt1* (the balance of local debt divided by the GDP of each region) is about 10%, and the maximum value is more than 50%, indicating large variations in the scale of local government debt across different regions. Both the mean and median of *GovDebt2* (the balance of local debt divided by the fiscal revenue of each region) exceed 150%, indicating that most provinces in China are under great pressure to repay local government debt.

## Main regression results

**Local government debt and enterprise tax burden.** Table 3 reports the impact of local government debt on corporate tax burden. Columns (1)-(3) are the regression results of the corporate income tax burden, and columns (4)-(6) are the regression results of the corporate comprehensive tax burden. The regression results of only controlling the year- and industry-fixed effects, adding company-level control variables, and adding macro-level control variables all show that the coefficient of local government debt is significantly positive above the 5% level. According to the regression results in columns (3) and (6), a 1% increase in the ratio of local government debt to GDP will increase the corporate income tax burden and the comprehensive tax burden of the jurisdiction by 4.6% and 5.4%, respectively. It shows that the larger the scale of local government debt, the higher the level of corporate tax burden within the jurisdiction. Hypothesis H1(a) is verified (the expansion of local government debt will increase the actual tax burden of enterprises). It preliminarily shows that when local governments face the fiscal pressure brought about by the expansion of local government debt, they will choose to "plunder" enterprises within their jurisdiction to increase tax revenue and ease debt pressure.

**Impact of different property rights.** Table 4 reports the regression results of the sample companies grouped by property rights. Columns (1) and (2) show that the expansion of the local government debt scale significantly increases the corporate income tax of non-state-owned enterprises at the 1% significance level, while having no significant impact on the actual income tax level of state-owned enterprises. Column (3) shows that the scale of local government debt increases the comprehensive tax burden of state-owned enterprises at the 5%

**Table 3. Local government debt and enterprise tax burden.**

| VARIABLES | (1) Taxburden1 | (2) Taxburden1 | (3) Taxburden1 | (4) Taxburden2 | (5) Taxburden2 | (6) Taxburden2 |
|---|---|---|---|---|---|---|
| GovDebt1 | 0.093*** | 0.047** | 0.046** | 0.032** | 0.054*** | 0.054*** |
| | (3.104) | (2.434) | (2.243) | (2.350) | (3.790) | (4.087) |
| Namerate | | 0.560*** | 0.560*** | | 0.116*** | 0.117*** |
| | | (13.541) | (13.506) | | (5.927) | (6.040) |
| Size | | -0.002 | -0.002 | | 0.002*** | 0.002*** |
| | | (-1.401) | (-1.438) | | (2.945) | (2.678) |
| Roa | | -0.610*** | -0.609*** | | 0.266*** | 0.268*** |
| | | (-11.485) | (-11.329) | | (11.463) | (11.840) |
| BM | | 0.014* | 0.014* | | -0.003 | -0.002 |
| | | (1.886) | (1.874) | | (-0.542) | (-0.443) |
| Lev | | 0.011 | 0.011 | | -0.050*** | -0.051*** |
| | | (1.294) | (1.307) | | (-11.257) | (-11.243) |
| Tangible | | 0.003 | 0.003 | | -0.007 | -0.008 |
| | | (0.291) | (0.268) | | (-1.328) | (-1.326) |
| Intangible | | 0.056* | 0.056* | | 0.058*** | 0.057*** |
| | | (1.740) | (1.720) | | (3.069) | (3.008) |
| Invent | | 0.034*** | 0.034*** | | -0.011 | -0.011 |
| | | (2.733) | (2.729) | | (-1.247) | (-1.169) |
| Loss | | 0.036*** | 0.036*** | | -0.004** | -0.004** |
| | | (4.396) | (4.392) | | (-1.972) | (-1.992) |
| Top10 | | -0.004 | -0.004 | | 0.010** | 0.010** |
| | | (-0.486) | (-0.463) | | (2.115) | (2.254) |
| Indep | | 0.050*** | 0.049*** | | 0.008 | 0.006 |
| | | (2.737) | (2.646) | | (0.597) | (0.475) |
| Density | | | -0.008 | | | -0.000** |
| | | | (-0.990) | | | (-2.119) |
| Inv | | | -0.056 | | | -0.129*** |
| | | | (-0.774) | | | (-3.491) |
| GDPr | | | 0.001 | | | 0.015 |
| | | | (0.045) | | | (1.316) |
| Constant | 0.108*** | 0.042 | 0.049 | 0.023*** | -0.049*** | -0.035* |
| | (5.095) | (1.113) | (1.302) | (3.026) | (-2.860) | (-1.840) |
| Year/Industry | YES | YES | YES | YES | YES | YES |
| Observations | 12,325 | 12,325 | 12,325 | 9,390 | 9,390 | 9,390 |
| Adj-R² | 0.141 | 0.261 | 0.261 | 0.172 | 0.296 | 0.299 |

Notes:

*, ** and *** indicate statistical significance at the 10, 5, and 1% levels, respectively. The values in parentheses are z-statistics and the standard errors are clustered at the provincial and municipal levels (31 groups) by using the wild cluster bootstrap method.

significance level, and column (4) shows that the scale of local government debt increases the comprehensive tax burden of non-state-owned enterprises at the 1% significance level. The regression results support hypothesis H2, implying that when local governments face the pressure of expanding debt, they are more inclined to "grab" the tax burden of non-state-owned enterprises to increase tax revenue and alleviate fiscal pressure.

**Impact of different tax authorities.** To further verify the role of local government intervention in the above research, this study divides the sample of enterprises into the State

**Table 4. Impact of different property rights.**

| VARIABLES | State-owned (1) Taxburden1 | Non-state-owned (2) Taxburden1 | State-owned (3) Taxburden2 | Non-state-owned (4) Taxburden2 |
|---|---|---|---|---|
| GovDebt1 | 0.014 | 0.058*** | 0.037** | 0.071*** |
| | (0.472) | (2.762) | (2.124) | (4.787) |
| Namerate | 0.479*** | 0.578*** | 0.090** | 0.119*** |
| | (6.643) | (12.188) | (2.458) | (6.791) |
| Size | -0.001 | -0.002 | 0.000 | 0.002* |
| | (-0.267) | (-0.995) | (0.249) | (1.765) |
| Roa | -0.963*** | -0.501*** | 0.406*** | 0.228*** |
| | (-9.838) | (-11.433) | (6.008) | (11.726) |
| BM | 0.004 | 0.018*** | 0.009 | -0.007* |
| | (0.224) | (2.629) | (0.894) | (-1.668) |
| Lev | 0.026 | -0.004 | -0.044*** | -0.048*** |
| | (1.335) | (-0.372) | (-4.468) | (-9.334) |
| Tangible | 0.025 | -0.010 | 0.007 | -0.022*** |
| | (1.326) | (-0.721) | (0.660) | (-3.825) |
| Intangible | 0.073* | 0.061 | 0.055** | 0.063*** |
| | (1.769) | (1.643) | (2.085) | (3.094) |
| Invent | 0.058** | 0.021 | 0.014 | -0.024** |
| | (2.194) | (1.524) | (0.946) | (-2.414) |
| Loss | 0.035*** | 0.036*** | -0.002 | -0.006** |
| | (2.585) | (3.285) | (-0.894) | (-2.034) |
| Top10 | -0.014 | 0.004 | 0.025*** | 0.002 |
| | (-0.877) | (0.364) | (3.134) | (0.352) |
| Indep | 0.091** | 0.030 | -0.003 | 0.010 |
| | (2.125) | (1.254) | (-0.151) | (0.897) |
| Density | -0.035* | 0.012 | -0.000 | -0.000* |
| | (-1.880) | (1.280) | (-1.583) | (-1.699) |
| Inv | -0.183 | 0.027 | -0.118*** | -0.137*** |
| | (-1.422) | (0.353) | (-2.868) | (-2.650) |
| GDPr | 0.028 | -0.016 | 0.020* | 0.009 |
| | (1.083) | (-0.603) | (1.723) | (0.581) |
| Constant | -0.011 | 0.071 | -0.024 | -0.019 |
| | (-0.166) | (1.486) | (-0.652) | (-0.856) |
| Year/Industry | YES | YES | YES | YES |
| Observations | 4,031 | 8,294 | 3,072 | 6,318 |
| Adj-$R^2$ | 0.265 | 0.236 | 0.372 | 0.260 |

Notes:

*, ** and *** indicate statistical significance at the 10, 5, and 1% levels, respectively. The values in parentheses are z-statistics and the standard errors are clustered at the provincial and municipal levels (31 groups) by using the wild cluster bootstrap method.

Taxation Bureau group and the Local Taxation Bureau group according to the different tax authorities.

Table 5 reports the impact of local government debt on the tax burden of enterprises collected and managed by different tax authorities. The results reveal that, among enterprises subject to the Local Taxation Bureau, the expansion of local government debt significantly

**Table 5. Impact of different tax authorities.**

| VARIABLES | Local Taxation Bureau (1) Taxburden1 | State Taxation Bureau (2) Taxburden1 | Local Taxation Bureau (3) Taxburden2 | State Taxation Bureau (4) Taxburden2 |
|---|---|---|---|---|
| GovDebt1 | 0.051*** | 0.042 | 0.046*** | 0.055*** |
|  | (2.695) | (1.354) | (2.889) | (4.214) |
| Namerate | 0.549*** | 0.574*** | 0.109*** | 0.124*** |
|  | (9.846) | (16.698) | (3.842) | (7.591) |
| Size | -0.001 | -0.004*** | 0.003** | 0.001 |
|  | (-0.266) | (-2.594) | (2.462) | (1.478) |
| Roa | -0.791*** | -0.447*** | 0.272*** | 0.260*** |
|  | (-10.337) | (-7.449) | (8.739) | (10.843) |
| BM | -0.001 | 0.024*** | -0.005 | -0.002 |
|  | (-0.086) | (2.661) | (-0.547) | (-0.341) |
| Lev | 0.003 | 0.022* | -0.061*** | -0.040*** |
|  | (0.164) | (1.855) | (-10.099) | (-8.589) |
| Tangible | 0.010 | -0.003 | -0.004 | -0.012 |
|  | (0.567) | (-0.220) | (-0.673) | (-1.521) |
| Intangible | 0.039 | 0.071** | 0.029 | 0.087*** |
|  | (0.841) | (2.115) | (1.562) | (3.286) |
| Invent | 0.057*** | 0.009 | 0.009 | -0.031*** |
|  | (2.761) | (0.498) | (0.768) | (-3.725) |
| Loss | 0.030*** | 0.042*** | -0.006* | -0.003 |
|  | (2.739) | (5.814) | (-1.653) | (-1.367) |
| Top10 | -0.001 | -0.002 | 0.015** | 0.007* |
|  | (-0.080) | (-0.188) | (2.025) | (1.744) |
| Indep | 0.056** | 0.049** | -0.002 | 0.015* |
|  | (2.031) | (2.410) | (-0.071) | (1.855) |
| Density | -0.009 | 0.005 | -0.000* | -0.000** |
|  | (-0.156) | (0.576) | (-1.686) | (-2.284) |
| Inv | -0.065 | -0.045 | -0.145*** | -0.119*** |
|  | (-0.657) | (-0.659) | (-2.975) | (-3.270) |
| GDPr | 0.001 | 0.017 | 0.006 | 0.029* |
|  | (0.018) | (0.684) | (0.400) | (1.819) |
| Constant | 0.036 | 0.031 | -0.046 | -0.021 |
|  | (0.620) | (1.016) | (-1.499) | (-1.374) |
| Year/Industry | YES | YES | YES | YES |
| Observations | 5,767 | 6,558 | 4,462 | 4,928 |
| Adj-$R^2$ | 0.243 | 0.276 | 0.296 | 0.310 |

Notes:

\*, \*\* and \*\*\* indicate statistical significance at the 10, 5, and 1% levels, respectively. The values in parentheses are z-statistics and the standard errors are clustered at the provincial and municipal levels (31 groups) by using the wild cluster bootstrap method.

increases their income tax burden, while for those subject to the State Taxation Bureau, the impact is not significant. The coefficients of Taxburden2 in the grouped regressions were slightly different, but all were significantly positive. These findings confirm hypothesis H3, which posits that the actual tax burden of enterprises is more sensitive to local government debt when they are managed by the Local Taxation Bureau. Therefore, the role of local government intervention in this impact is further confirmed.

### Robustness tests

**Endogeneity discussion.** The increase in the regional tax revenue may give local governments the confidence to repay their debts, which in turn encourages local governments to increase borrowing within the bond issuance limit. To avoid the endogeneity problem of mutual causality, we adopt the instrumental variable method and lagged explanatory variables to address the issue of endogeneity.

First, drawing on the existing research [9, 48], we use the mean value of education expenditure of other provinces and cities in the same year as the instrumental variable (*Espend*). On the one hand, local officials' assessment in China includes the inspection of people's livelihood matters, so there is competition among local governments in the scale of fiscal expenditure on people's livelihood [49]. That is, when other regions increase fiscal expenditure to improve people's livelihood performance, local governments will increase fiscal expenditure accordingly for regional competition. Moreover, there is a certain correlation between government revenue and expenditure and the scale of local government debt, which meets the correlation requirements of instrumental variables. On the other hand, education expenditure is the basic expenditure of local finance on people's livelihood, which belongs to the local rigid expenditure and has a weak correlation with the short-term economic fluctuations that affect the actual tax burden of enterprises. This means *Espend* is relatively exogenous. At the same time, the pressure of fiscal expenditure in other regions will not be directly transferred to enterprises within the jurisdiction. Considering these features, *Espend* meets the exogeneity requirements of instrumental variables.

Table 6 reports the regression results of the instrumental variables in Columns (1)-(4). The instrumental variable (*Espend*) passed the underidentification test and the weak identification test. The regression results in columns (1) and (3) show that *Espend* is positively correlated with *GovDebt1* at the significance level of 1%. And the regression results in columns (2) and (4) show that after the endogeneity problem is controlled, the impact of local government debt on the actual tax burden of enterprises is still significantly positive at the level of 5%, indicating that the results of this paper are robust to a certain extent.

Second, to exclude the stimulating effect of the current tax revenue increase on local bond issuance, we lag the main explanatory variable (*GovDebt1*) by one period. Columns (5) and (6) show that the regression results are essentially consistent with the previous ones, providing further evidence of the robustness of the regression results.

**Robustness tests.** First, Dalian, Qingdao, Ningbo, Xiamen, and Shenzhen are Chinese Independent Plan Cities, and their taxes are independently collected and managed by municipal departments. So, we match the main explanatory variable local government debt scale (*GovDebt1*) to 36 regions (31 provinces, municipalities, autonomous regions and 5 independent plan cities), and carry out the regression again. Column (1) and (2) in Table 7 reports the regression results based on 36 regions. After adjusting the sample matching range, the scale of local government debt still significantly increases the actual tax burden of enterprises.

Second, to ensure the reliability of the results, this paper selects the local government debt ratio (*GovDebt2*) as a proxy indicator of the debt scale to re-regress. Column (3) and (4) in Table 9 reports the regression results that the scale of local government debt still increases the actual tax burden of enterprises after replacing the indicators.

## Additional analysis

### Path testing: Tax collection effort and tax incentive intensity

Although China's statutory tax rates are set uniformly by the central government, numerous studies have shown that local governments have certain discretion in the process of tax

**Table 6. Address endogeneity problems.**

| Variables | (1) first GovDebt1 | (2) second Taxburden1 | (3) first GovDebt1 | (4) second Taxburden2 | (5) Taxburden1 | (6) Taxburden2 |
|---|---|---|---|---|---|---|
| Espend | 0.002*** | | 0.002*** | | | |
| | (7.47) | | (7.67) | | | |
| GovDebt1 | | 0.0766*** | | 0.0625** | | |
| | | (3.98) | | (2.98) | | |
| GovDebt1_1 | | | | | 0.057*** | 0.053*** |
| | | | | | (2.875) | (3.334) |
| Namerate | 0.029 | 0.559*** | 0.003 | 0.117*** | 0.549*** | 0.105*** |
| | (0.67) | (13.68) | (0.10) | (6.14) | (12.530) | (4.680) |
| Size | 0.000 | -0.002 | -0.001 | 0.002** | -0.003* | 0.002** |
| | (-0.09) | (-1.51) | (-0.68) | (2.72) | (-1.697) | (2.489) |
| Roa | -0.053 | -0.608*** | -0.034 | 0.268*** | -0.582*** | 0.242*** |
| | (-1.41) | (-11.63) | (-0.97) | (11.97) | (-12.196) | (8.715) |
| BM | -0.004 | 0.014 | 0.007 | -0.002 | 0.008 | -0.008 |
| | (-0.60) | (1.91) | (0.97) | (-0.49) | (0.884) | (-1.115) |
| Lev | 0.010 | 0.011 | 0.010 | -0.051*** | 0.014 | -0.048*** |
| | (1.13) | (1.32) | (1.17) | (-11.61) | (1.325) | (-7.138) |
| Tangible | 0.006 | 0.002 | 0.008 | -0.008 | 0.004 | -0.007 |
| | (0.54) | (0.18) | (0.68) | (-1.38) | (0.336) | (-1.113) |
| Intangible | -0.023 | 0.056 | -0.020 | 0.057** | 0.093*** | 0.062** |
| | (-0.73) | (1.71) | (-0.71) | (3.08) | (2.847) | (2.465) |
| Invent | -0.021 | 0.034** | -0.022 | -0.011 | 0.027* | -0.022** |
| | (-1.20) | (2.71) | (-1.34) | (-1.20) | (1.704) | (-1.985) |
| Loss | 0.004 | 0.036*** | 0.005 | -0.004* | - | 0.010 |
| | (1.49) | (4.43) | (1.72) | (-2.11) | | (1.122) |
| Top10 | -0.014 | -0.003 | -0.009 | 0.010* | -0.012 | 0.010** |
| | (-1.75) | (-0.39) | (-0.91) | (2.25) | (-1.508) | (1.977) |
| Indep | 0.025 | 0.049** | 0.016 | 0.006 | 0.055*** | 0.005 |
| | (0.79) | (2.74) | (0.68) | (0.47) | (3.776) | (0.348) |
| Density | -0.273*** | -0.002 | 0.000*** | 0.000 | -0.008 | -0.000* |
| | (-3.66) | (-0.30) | (-3.74) | (-1.88) | (-0.856) | (-1.649) |
| Inv | 0.208 | -0.058 | 0.132 | -0.130*** | -0.054 | -0.127*** |
| | (0.58) | (-0.83) | (0.38) | (-3.61) | (-0.702) | (-3.186) |
| GDPr | -0.345** | 0.014 | -0.354** | 0.018 | 0.006 | 0.015 |
| | (-3.05) | (0.67) | (-3.12) | (1.35) | (0.257) | (1.453) |
| Constant | -1.531*** | 0.041 | -1.495*** | -0.037* | 0.065* | -0.038* |
| | (-6.27) | (1.13) | (-6.69) | (-1.99) | (1.956) | (-1.647) |
| Year/Industry | YES | YES | YES | YES | YES | YES |
| Observations | 12,325 | 12,325 | 9,390 | 9,390 | 8,437 | 5,706 |
| $R^2$ | | 0.263 | | 0.302 | | |
| Adj-$R^2$ | 0.536 | | 0.564 | | 0.264 | 0.288 |
| Underidentification test: Kleibergen-Paap rk LM statistic | 5.055** | | 5.275** | | | |
| Weak identification test: Kleibergen-Paap Wald rk F statistic | 55.84*** | | 58.84*** | | | |

(*Continued*)

**Table 6.** (Continued)

| | (1) | (2) | (3) | (4) | (5) | (6) |
|---|---|---|---|---|---|---|
| | *first* | *second* | *first* | *second* | | |
| *Variables* | *GovDebt1* | *Taxburden1* | *GovDebt1* | *Taxburden2* | *Taxburden1* | *Taxburden2* |
| *Overidentification test of all instruments: Hansen J statistic* | equation exactly identified | | equation exactly identified | | | |

Notes:

\*, \*\* and \*\*\* indicate statistical significance at the 10, 5, and 1% levels, respectively. The values in parentheses in columns (1) and (3) are t-statistics, and the values in parentheses in columns (2), (4), (5) and (6) are z-statistics. The standard errors are clustered at the provincial and municipal levels (31 groups) by using the wild cluster bootstrap method.

collection [29, 31]. Since the taxation department's collection work requires the support and cooperation of the local government, the collection intensity of the taxation authority will also be affected by the local government [13]. Furthermore, China banned local governments from providing illegal tax rebates to enterprises in 2002, and announced the cleanup and standardization of local preferential tax policies in 2014, indicating that local governments have always had a say in the formulating and implementing of preferential tax policies. In 2017, the "Notice of the State Council on Several Measures for Expanding Opening-up and Actively Utilizing Foreign Investment" set up a transition period for cleaning up the original tax incentives, which allowed local governments to formulate preferential tax policies within the scope of legal authority. Based on the above, we believe that when facing the fiscal needs brought about by local debt pressure, local governments will implement tax intervention by strengthening tax collection effort and reducing tax incentives, in order to increase tax revenue in the short term and meet the needs of debt repayment and rigid expenditure.

Drawing on the estimation of regional tax collection effort from existing research [14, 53, 54], we construct the tax collection effort (*TE*) indicator [14, 50, 51]. Specifically, Eq (2) is used to estimate the expected tax revenue of each region, and *TE* is the ratio of the actual tax revenue to the estimated tax revenue of each region. A higher value of *TE* indicates a stronger tax collection intensity in that region. At the same time, referring to the research of Liu [53], we estimate the tax incentive intensity (*Incentive*) accepted by enterprises. Specifically, tax incentive intensity = various tax rebates received/(various tax rebates received + various taxes paid). Then, we examine the mediating effect of regional tax collection effort and tax incentive intensity.

$$\frac{TAX_{j,t}}{GDP_{j,t}} = \theta_0 + \theta_1 \times \frac{IND1_{j,t}}{GDP_{j,t}} + \theta_2 \times \frac{IND2_{j,t}}{GDP_{j,t}} + \theta_3 \times \frac{IND3_{j,t}}{GDP_{j,t}} + \theta_4 \times \frac{OPENNESS_{j,t}}{GDP_{j,t}} + \varepsilon \quad (2)$$

Where *j* represents the region, *t* represents the year, *TAX* is the actual tax revenue of each province at the end of the year, $\frac{TAX_{j,t}}{GDP_{j,t}}$ is the year-end tax revenue of each region divided by GDP, *IND1*, *IND2*, *IND3* is the output value of the primary, secondary, and tertiary industries of each region, respectively, and *OPENNESS* represents the openness of the region, which is the year-end import and export value of each region. The ratio of the actual tax burden ratio ($\frac{TAX_{j,t}}{GDP_{j,t}}$) to the expected tax burden ratio ($\frac{TAX_{j,t}}{GDP_{j,t}}'$) is used to represent the intensity of tax collection, and a higher ratio indicates a higher intensity of tax collection in the region.

Columns (1)-(4) in Table 8 report the impact of local government debt on regional tax collection effort, and the role that regional tax collection effort plays in the impact of local government debt on the tax burden of enterprises. Among them, columns (1) and (2) are corporate income tax. In column (1), the coefficient of GovDebt1 is significantly positive at the level of

**Table 7. Robustness tests.**

| Variables | (1) Taxburden1 | (2) Taxburden1 | (3) Taxburden2 | (4) Taxburden2 |
|---|---|---|---|---|
| GovDebt1_36 | 0.049*** | 0.066*** | | |
| | (2.635) | (5.250) | | |
| GovDebt2 | | | 0.004** | 0.005*** |
| | | | (1.976) | (3.193) |
| Namerate | 0.565*** | 0.116*** | 0.560*** | 0.117*** |
| | (13.735) | (6.018) | (13.514) | (5.863) |
| Size | -0.002 | 0.002*** | -0.002 | 0.002*** |
| | (-1.468) | (2.846) | (-1.331) | (2.868) |
| Roa | -0.611*** | 0.268*** | -0.612*** | 0.265*** |
| | (-11.498) | (11.799) | (-11.234) | (11.845) |
| BM | 0.014* | -0.003 | 0.014* | -0.003 |
| | (1.843) | (-0.514) | (1.789) | (-0.536) |
| Lev | 0.013 | -0.051*** | 0.011 | -0.051*** |
| | (1.447) | (-11.087) | (1.277) | (-11.301) |
| Tangible | 0.002 | -0.008 | 0.003 | -0.008 |
| | (0.150) | (-1.277) | (0.270) | (-1.329) |
| Intangible | 0.054* | 0.058*** | 0.055* | 0.058*** |
| | (1.651) | (3.057) | (1.739) | (2.999) |
| Invent | 0.034*** | -0.011 | 0.033*** | -0.011 |
| | (2.710) | (-1.182) | (2.655) | (-1.230) |
| Loss | 0.036*** | -0.005** | 0.037*** | -0.004** |
| | (4.367) | (-2.291) | (4.424) | (-1.967) |
| Top10 | -0.004 | 0.010** | -0.004 | 0.010** |
| | (-0.441) | (2.100) | (-0.487) | (2.213) |
| Indep | 0.050*** | 0.008 | 0.049*** | 0.007 |
| | (2.729) | (0.574) | (2.642) | (0.526) |
| Density | -0.000 | -0.000** | -0.002 | -0.000 |
| | (-1.513) | (-2.232) | (-0.154) | (-0.695) |
| Inv | -0.045 | -0.110*** | -0.055 | -0.128*** |
| | (-0.916) | (-4.685) | (-0.699) | (-3.127) |
| GDPr | -0.004 | 0.010 | -0.007 | 0.006 |
| | (-0.189) | (0.982) | (-0.326) | (0.452) |
| Constant | 0.051 | -0.042** | 0.049 | -0.036* |
| | (1.280) | (-2.353) | (1.300) | (-1.867) |
| Year/Industry | YES | YES | YES | YES |
| Observations | 12,303 | 9,367 | 12,325 | 9,390 |
| Adj-$R^2$ | 0.262 | 0.300 | 0.261 | 0.297 |

Notes:

*, ** and *** indicate statistical significance at the 10, 5, and 1% levels, respectively. The values in parentheses are z-statistics and the standard errors are clustered at the provincial and municipal levels (31 groups) by using the wild cluster bootstrap method.

1%, indicating that the expansion of the local government debt scale significantly increases the intensity of regional tax collection. The coefficient of TE in column (2) is significantly positive at the level of 5%, which plays an incomplete intermediary effect, indicating that the fiscal pressure brought about by the expansion of local government debt makes the local government intervene in regional tax collection behavior. By increasing the intensity of tax collection and

**Table 8. Path testing: Tax collection effort and tax incentive intensity.**

| VARIABLES | (1) TE | (2) Taxburden1 | (3) TE | (4) Taxburden2 | (5) Incentive | (6) Taxburden1 | (7) Incentive | (8) Taxburden2 |
|---|---|---|---|---|---|---|---|---|
| GovDebt1 | 0.048*** | 0.033 | 0.050*** | 0.047*** | -0.356*** | 0.016 | -0.179*** | 0.029*** |
| | (3.744) | (1.557) | (4.245) | (3.090) | (-7.795) | (0.805) | (-4.808) | (3.065) |
| TE | | 0.270** | | 0.154* | | | | |
| | | (2.242) | | (1.671) | | | | |
| Incentive | | | | | | -0.046*** | | -0.140*** |
| | | | | | | (-6.226) | | (-35.956) |
| Namerate | 0.009 | 0.558*** | 0.006 | 0.116*** | -0.327*** | 0.522*** | -0.326*** | 0.071*** |
| | (1.575) | (13.552) | (0.936) | (5.929) | (-4.548) | (11.418) | (-6.320) | (4.698) |
| Size | 0.000 | -0.002 | 0.000 | 0.002** | -0.006* | -0.003** | 0.002 | 0.002*** |
| | (0.270) | (-1.467) | (0.277) | (2.544) | (-1.689) | (-2.069) | (0.914) | (3.766) |
| Roa | 0.006 | -0.611*** | 0.001 | 0.268*** | -0.470*** | -0.630*** | -0.265*** | 0.231*** |
| | (1.124) | (-11.255) | (0.225) | (11.912) | (-7.546) | (-10.468) | (-6.367) | (11.266) |
| BM | 0.002 | 0.013* | 0.003* | -0.003 | -0.044* | 0.017** | -0.032** | -0.007 |
| | (1.571) | (1.753) | (1.891) | (-0.542) | (-1.891) | (2.327) | (-2.224) | (-1.501) |
| Lev | 0.001 | 0.011 | 0.001 | -0.051*** | 0.055** | 0.023*** | 0.001 | -0.050*** |
| | (0.679) | (1.262) | (0.360) | (-11.239) | (2.196) | (2.612) | (0.070) | (-12.826) |
| Tangible | 0.006*** | 0.002 | 0.006*** | -0.008 | 0.009 | 0.004 | -0.044** | -0.014** |
| | (3.847) | (0.131) | (3.004) | (-1.514) | (0.442) | (0.341) | (-2.070) | (-2.463) |
| Intangible | 0.005 | 0.054* | 0.004 | 0.057*** | -0.231*** | 0.051 | -0.114*** | 0.041** |
| | (1.147) | (1.658) | (0.778) | (3.014) | (-3.141) | (1.470) | (-2.905) | (2.327) |
| Invent | 0.002 | 0.034*** | 0.001 | -0.011 | -0.008 | 0.029*** | 0.021 | -0.008 |
| | (1.374) | (2.660) | (0.897) | (-1.187) | (-0.163) | (2.926) | (0.753) | (-0.934) |
| Loss | -0.000 | 0.036*** | 0.000 | -0.004** | 0.009 | 0.036*** | 0.001 | -0.004** |
| | (-0.457) | (4.431) | (0.511) | (-2.016) | (0.676) | (3.654) | (0.188) | (-2.279) |
| Top10 | -0.004*** | -0.003 | -0.004** | 0.011** | -0.002 | -0.001 | -0.043*** | 0.004 |
| | (-2.913) | (-0.319) | (-2.523) | (2.307) | (-0.086) | (-0.061) | (-2.924) | (1.085) |
| Indep | -0.003 | 0.050*** | -0.002 | 0.007 | 0.107 | 0.053** | 0.058 | 0.014 |
| | (-1.088) | (2.651) | (-0.769) | (0.502) | (1.403) | (2.266) | (1.448) | (1.431) |
| Density | 0.012 | -0.011 | 0.000 | -0.000** | 0.033 | -0.003 | 0.000 | -0.000*** |
| | (1.353) | (-1.260) | (1.388) | (-2.481) | (1.079) | (-0.391) | (0.590) | (-2.908) |
| Inv | 0.035 | -0.065 | 0.047 | -0.137*** | 0.483 | -0.073 | 0.345** | -0.081*** |
| | (0.486) | (-0.867) | (0.605) | (-3.870) | (1.598) | (-1.198) | (2.053) | (-3.665) |
| GDPr | -0.046*** | 0.014 | -0.049*** | 0.022** | -0.142*** | -0.011 | -0.081** | 0.003 |
| | (-2.746) | (0.587) | (-2.668) | (1.974) | (-2.660) | (-0.562) | (-2.395) | (0.385) |
| Constant | 0.029*** | 0.041 | 0.027** | -0.039** | 0.458*** | 0.089* | 0.181** | -0.010 |
| | (2.905) | (1.071) | (2.344) | (-2.119) | (4.768) | (1.919) | (2.314) | (-0.585) |
| Year/Industry | YES | YES | YES | YES | YES | YES | YES | YES |
| Observations | 12,325 | 12,325 | 9,390 | 9,390 | 9,838 | 9,838 | 9,390 | 9,390 |
| Adj-R² | 0.289 | 0.261 | 0.282 | 0.300 | 0.110 | 0.255 | 0.109 | 0.438 |

Notes:

*, ** and *** indicate statistical significance at the 10, 5, and 1% levels, respectively. The values in parentheses are z-statistics and the standard errors are clustered at the provincial and municipal levels (31 groups) by using the wild cluster bootstrap method.

increasing the actual tax burden of enterprises, the local government can achieve the purpose of increasing fiscal revenue and alleviating debt pressure. The regression results of the comprehensive tax burden of enterprises in columns (3) and (4) are basically consistent with those in

columns (1) and (2). Columns (5)-(8) report the impact of local government debt on the intensity of tax incentives, and the role of tax incentives in the impact of local government debt on corporate tax burdens. Columns (5) and (7) show that with the expansion of local government debt, the level of tax incentives received by enterprises in the region will decrease. The results in columns (6) and (8) show that under the background of local debt expansion, the fewer tax incentives, the higher the actual tax burden of enterprises, and the mediating effect of tax incentives are established. The regression results of the two different intervention methods in Table 6 reveal the mechanism of local debt expansion to increase the actual tax burden of enterprises, which further verifies local government intervention in this process.

## Heterogeneity test of the institutional environment

China's market-oriented reform is still in progress [52], and the institutional environment and the attractiveness of the mobile tax base of different regions are quite different. As the price paid by enterprises to obtain regional rights and public services, the enterprise tax burden is the object that enterprises weigh with other available resources in the process of investment decision-making [53]. A good institutional environment builds an active competitive atmosphere and trading market, which can provide enterprises with a better business environment and better public products [54]. It improves the tolerance of enterprises to the actual tax burden level and is an effective substitute for the local government to give enterprises a relaxed taxation environment, while regions with a poor institutional environment are difficult to provide an effective guarantee for business transactions, and their economic attractiveness and market competitiveness are low. In such regions, if the tax collection intensity is increased and the tax incentive intensity is reduced, the tax base of the jurisdiction may be lost. Therefore, we believe that there will be heterogeneity in the tax interference of local governments in different institutional environments. Specifically, regions with a better institutional environment have higher attractiveness for mobile enterprises, and the pressure to provide a relaxed tax collection environment for enterprises is weaker. Facing the expansion of the debt scale, they tend to adjust the tax collection intensity upward and the tax incentive intensity downward, so as to increase the actual tax burden of enterprises and alleviate the debt pressure in the short term, while in regions with a poor institutional environment, other resources for economic development are relatively scarce. They cannot provide an active trading environment and sufficient investment protection for enterprises in the jurisdiction. The attraction of the liquidity tax base is weak. Therefore, when considering the implementation of stricter tax collection, such regions must first consider the issue of "escape" of the liquid tax base. After weighing the current fiscal pressure and future regional development, they may choose to "plan for the long term", and tend to provide a relatively relaxed tax collection environment to enhance the economic attractiveness of the jurisdiction.

Given the above, to explore whether there are differences in local government intervention in different institutional environment regions, this study conducts a group test. Based on the existing research, the institutional environment index is constructed from the three dimensions of the market mechanism, government governance and the level of rule of law [21]. Specifically, the index in *China's Provincial Marketization Index Report (2018)* [52] is used as the basic data. Among them, the "Regional marketization process index" measures the level of regional marketization, the "Index regarding the government's reduction of its intervention in firms" measures the level of government governance, and the "Development of market intermediary organizations and legal environment index" measures the level of regional rule of law. If the mean value of the three is higher than the tertile of the indicator, it is classified as the group with a better institutional environment; otherwise, it is the group with a poor institutional environment.

**Table 9. Heterogeneity test of the institutional environment.**

*Panel A*: *Taxburden1*

| | better institutional environment | | | poor institutional environment | | |
|---|---|---|---|---|---|---|
| | (1) | (2) | (3) | (4) | (5) | (6) |
| VARIABLES | Taxburden1 | TE | Incentive | Taxburden1 | TE | Incentive |
| GovDebt1 | 0.086*** | 0.043** | -0.216*** | 0.012 | 0.026 | -0.069* |
| | (5.834) | (2.048) | (-5.296) | (0.337) | (1.301) | (-1.851) |
| Constant | 0.057 | 0.037*** | 0.459*** | 0.021 | 0.018 | 0.192 |
| | (1.293) | (3.554) | (4.008) | (0.301) | (1.239) | (1.029) |
| Controls | Yes | Yes | Yes | Yes | Yes | Yes |
| Observations | 9,192 | 9,192 | 7,589 | 3,133 | 3,133 | 2,249 |
| Adj-$R^2$ | 0.280 | 0.300 | 0.103 | 0.224 | 0.369 | 0.123 |

*Panel B*: *Taxburden2*

| | better institutional environment | | | poor institutional environment | | |
|---|---|---|---|---|---|---|
| | (1) | (2) | (3) | (4) | (5) | (6) |
| VARIABLES | Taxburden2 | TE | Incentive | Taxburden2 | TE | Incentive |
| GovDebt1 | 0.040* | 0.063** | -0.156** | 0.009 | 0.017 | -0.024 |
| | (1.716) | (2.414) | (-2.180) | (0.413) | (1.156) | (-0.698) |
| Constant | -0.058*** | 0.025* | 0.215** | 0.081** | 0.033* | -0.075 |
| | (-2.798) | (1.698) | (2.370) | (2.016) | (1.944) | (-0.602) |
| Controls | Yes | Yes | Yes | Yes | Yes | Yes |
| Observations | 7,098 | 7,098 | 7,098 | 2,292 | 2,292 | 2,292 |
| Adj-$R^2$ | 0.299 | 0.264 | 0.102 | 0.348 | 0.213 | 0.144 |

Notes: *, ** and *** indicate statistical significance at the 10, 5, and 1% levels, respectively. The values in parentheses are z-statistics and the standard errors are clustered at the provincial and municipal levels (31 groups) by using the wild cluster bootstrap method.

The grouping results in Table 9 show that under different institutional environments, there are significant differences in the impact of local government debt on the actual tax burden of enterprises. There are also significant differences in the intervention of local governments. Specifically, the regression results of columns (1)-(3) are consistent with the previous. In regions with a better institutional environment, the expanded debt scale makes local governments increase the tax collection intensity and reduce tax incentive intensity, thereby significantly increasing the actual tax burden of enterprises in their jurisdictions. However, the regression results of columns (4)-(6) show that when local governments in areas with a poor institutional environment are facing debt repayment pressure, they do not choose stricter tax collection behaviors. As a result, the actual tax burden of enterprises in such regions has not increased. The results of this grouping test show that, due to the low market competitiveness and low capital attraction, regions with poor institutional environment need to choose between short-term tax growth and long-term economic growth. Even if local governments bear heavy debt pressure, to avoid other losses caused by the escape of regional tax base, these regions still choose to "take a long-term view". That is to provide a loose tax collection environment for enterprises in the jurisdiction, and realize the long-term development of the regional economy by reducing the tax burden of enterprises and stabilizing the tax base.

## Conclusion

The solution to existing risks and the prevention of incremental risks of local government debt will still be the main line of China's future policies. Especially in the context of the continuous

recurrence of the global COVID-19 epidemic and many uncertain factors in the process of economic recovery, China's task of preventing and resolving major risks caused by local debt remains arduous. Starting from the motivation and method of local government intervention in tax collection and management, this study investigates the impact of local government debt on the actual tax burden of enterprises, and further explores the path of the impact. The results of the study show that, in general, the expansion of local government debt has increased the actual tax burden of enterprises, which is mainly reflected in non-state-owned enterprises and enterprises that are collected and managed by the Local Taxation Bureau. The path test shows that local governments affected the actual tax burden of enterprises in their jurisdiction by adjusting the intensity of tax collection and tax incentives. Further testing for heterogeneity shows that when local governments faced debt servicing pressure, local governments would make a trade-off between "short-term tax revenue" and "long-term tax revenue". Specifically, regions with a better institutional environment are more attractive to the liquid tax base. Local governments in such regions tend to increase the tax collection intensity and reduce tax incentive intensity. And they will ease debt repayment pressure by increasing the actual tax burden of enterprises. However, due to the lack of market competitiveness in the regions with a poor institutional environment, they tend to provide a more relaxed tax collection environment to stabilize the tax base and choose a "long-term plan" to resolve debts.

The research conclusions of this paper have the following policy implications: First, China needs to further regulate the local government's borrowing and financing, and improve the local government's borrowing mechanism by the law. Establish a market-oriented and legalized debt default disposal mechanism to continuously manage and control the risk of local government debt. Second, the institutional environment plays an important role in economic growth and regional competition. Therefore, local governments should strive to improve the institutional environment of the region, improve the level of marketization, legalization and internationalization of the business environment, and enhance the regional competitive advantages, so as to strengthen the attraction and allocation of resources. Third, continue to deepen the reform of the fiscal and taxation system, and accelerate the establishment of a modern fiscal and taxation system that is conducive to high-quality development. Further, rationalize the fiscal relationship between the central and local governments. Under the premise of unified legislation by the central government, the authority of local tax administration should be appropriately expanded through legislative authorization, and the functions of local governments should be optimized. Reduce the mismatch between financial rights, administrative powers and expenditure responsibilities, form a balanced and stable financial system that is commensurate with financial resources, and provide guarantees for local governments to improve high-quality public resources. At the same time, establish and improve the local tax system, optimize the tax system structure, and cultivate tax sources to ensure sustainable local fiscal revenue. Fourth, deepen the reform of the tax collection and administration system. Continue to promote the Smart Taxation system driven by tax big data, and improve the tax supervision system. Cultivate a market-oriented, legalized, and international business environment, create a fair and efficient tax payment environment for enterprises, and enhance enterprises' satisfaction with tax reduction.

Taking the actual tax burden level of micro-enterprises as the research object, this paper expands the research on the economic consequences of local governments' expanding bond issuance in developing countries. At the same time, through mechanism test, heterogeneity analysis, and other further tests, this paper reveals the role of local governments' intervention behavior in this process, analyzes the trade-off of local governments' taxation under different institutional environments, and enriches the literature in tax collection, regional competition, and other related fields. It provides policy enlightenment for developing countries with large

differences in regional development levels to improve the public debt management system, create a fair tax environment, and promote high-quality economic growth, which has important theoretical and practical significance. However, due to the limitations of data and research conditions, there are still some problems that need to be further explored. First of all, due to the lack of data, the regional tax collection intensity index in this paper refers to the estimation method of the existing research, which lacks the examination of the changes in the actual tax collection behavior of the tax authorities. If we can directly describe the intensity of tax collection through the tax authorities' behaviors (such as strengthening pre-supervision, in-process investigation and post-inspection), we can more accurately and carefully observe the impact of local government debt expansion on regional tax collection and administration, and put forward corresponding policy suggestions. Second, the research conclusion of this paper shows that local governments will intervene in tax collection when faced with expanding debt scale, which will increase the tax burden of enterprises within their jurisdiction. However, due to the lack of data, the analysis has not been carried out from other perspectives, such as whether local governments support industries with higher tax contributions in order to obtain more tax revenue, and how underdeveloped regions that do not obtain fiscal revenue growth by intervening in the tax collection deal with the pressure of debt expansion in the short term. The above issues still need further discussion in the future.

## Author Contributions

**Conceptualization:** Wei Tang.

**Data curation:** Xingzhu Zhao.

**Funding acquisition:** Wei Tang, Xingzhu Zhao.

**Methodology:** Xingzhu Zhao.

**Project administration:** Shengbao Zhai.

**Supervision:** Wei Tang, Shengbao Zhai.

**Visualization:** Shengbao Zhai.

**Writing – original draft:** Xingzhu Zhao.

**Writing – review & editing:** Wei Tang, Lei Cao.

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
