## [Decision Letter · Decision Letter 0]

6 Dec 2022

PONE-D-22-28623Local government debt and corporate tax burden: a perspective based on the trade-off of government tax collection and managementPLOS ONE

Dear Dr. Zhao,

Thank you for submitting your manuscript to PLOS ONE. After careful consideration, we feel that it has merit but does not fully meet PLOS ONE’s publication criteria as it currently stands. Therefore, we invite you to submit a revised version of the manuscript that addresses the points raised during the review process. Please submit your revised manuscript by Jan 20 2023 11:59PM. If you will need more time than this to complete your revisions, please reply to this message or contact the journal office at plosone@plos.org. Please include the following items when submitting your revised manuscript:A rebuttal letter that responds to each point raised by the academic editor and reviewer(s). You should upload this letter as a separate file labeled 'Response to Reviewers'.A marked-up copy of your manuscript that highlights changes made to the original version. You should upload this as a separate file labeled 'Revised Manuscript with Track Changes'.An unmarked version of your revised paper without tracked changes. You should upload this as a separate file labeled 'Manuscript'.

We look forward to receiving your revised manuscript.

Kind regards,

Dr. Florian Follert

Academic Editor

PLOS ONE

Journal Requirements:

   "Initials of the authors who received each award: T

Grant numbers awarded to each author: 20BJY023

The full name of each funder: National Social Science Foundation of China 

URL of each funder website: NO

Did the sponsors or funders play any role in the study design, data collection and analysis, decision to publish, or preparation of the manuscript?  NO

Initials of the authors who received each award: Z

Grant numbers awarded to each author: ACYC2020376

The full name of each funder: Anhui University of Finance and Economics Master's Degree Innovation Fund Project  

URL of each funder website: NO

Did the sponsors or funders play any role in the study design, data collection and analysis, decision to publish, or preparation of the manuscript?  NO"

Additional Editor Comments:

I have carefully read both the paper and the referee reports. Both reviewers see current weaknesses in the paper, most of them I share. However, the reviewers give valuable hints to improve the paper. Therefore, I am giving the authors the opportunity to revise the manuscript. I expect that all aspects of reviewer 2 will be implemented in the revision. Reviewer 1 recommends to reject the paper. The authors should please respond to each of reviewer 1's points and, if possible, try to implement the suggestions. In particular, those aspects concerning the rigour of the analysis must be addressed. 

Reviewers' comments:

Reviewer's Responses to Questions

**Comments to the Author**

1. Is the manuscript technically sound, and do the data support the conclusions?

Reviewer #1: No

Reviewer #2: Yes

2. Has the statistical analysis been performed appropriately and rigorously? 

Reviewer #1: No

Reviewer #2: Yes

3. Have the authors made all data underlying the findings in their manuscript fully available?

Reviewer #1: Yes

Reviewer #2: No

4. Is the manuscript presented in an intelligible fashion and written in standard English?

Reviewer #1: No

Reviewer #2: Yes

5. Review Comments to the Author

Reviewer #1: Review of „Local government debt and corporate tax burden”

The paper looks at the relationship between the development of public debt on regional level in China and the efforts to collect more taxes from corporations. The hypothesis tested is whether the municipalities have incentives to increase their collection efforts as a result in increases in public debt to reduce the fiscal pressure. The authors find, using a panel regression with fixed effects and an instrumental variables specification, that higher levels of public debt correlate positively with the tax burden of companies.

While the topic is interesting and relevant, the methodology of investigation is weak and does not support the conclusions. Moreover, the results are very context specific, which raises doubts regarding the external validity and thus interest for a general-interest journal such as Plos One. I do not recommend publication of the paper.

Major comments:

• Panel estimation: while the number of observations looks impressive, as the authors treat year-companies as a unit of observation, closer look reveals that the actual variation is much lower. Firstly, only five years are considered, which limits the time variation of the main explanatory variable. Secondly, and more importantly, local public debt is measures at the regional (local?) level (in fact, nowhere in the paper it is specified what the actual unit of observation of the debt variable is!). Implicitly, it seems that this is provincial data (e.g., referring to other provinces in the construction of the instrument), which means that the spatial variation is at 23 provinces or 34 regions. This does not allow for serious panel estimation. Moreover, this estimation requires an inclusion of regional fixed effects – which are not included. Why not?

• Related to the previous point is the question of endogeneity. An obvious candidate for a confounder is the overall economic activity in a region, which affects both the denominator of the debt ratio and (indirectly) the denominator of the tax burden, e.g., through operating income or pre-tax profit. This means, that any changes to the economic situation of each province, will likely affect both sides of the regression equation. Authors use in one specification a Hausman-type spatial instrument, which replaces own debt levels, with an average of debt levels of other provinces. This instrument does not resolve the endogeneity issue, as any idiosyncratic change to the economic situation of all regions still affects both the explanatory and the outcome variables. Besides, no test statistics of the instruments, such as first-stage regressions, or the overidentification tests, F-Statistics and like to assess the quality of the instrument are provided.

• The overall empirical exercise is very context specific. The authors mention, that the channel of transmission for the higher tax burden is essentially only the tax collection effort, as the taxation rates and bases as such as regulated nation-wide. This raises the question of external validity – what can we learn from the paper on the similar cases all over the world?

• Hypotheses 2 and 3 have left me a bit puzzled. The authors suggest that the impact of debt on tax burden should be *more significant* in non-state-owned enterprises. I would think that there should *only* be an effect in those companies, unless there is something about China, which makes is “normal” that state-owned entreprises do not fulfill their tax obligations. If this is indeed the case, it needs to be explained in more detail.

Minor comments:

• Page 3, line 38: what are financing platforms?

• Pge 8, line 155: what is land finance? Does this refer to management of municipal property?

• I don’t understand the difference between the local and state taxation bureaus and their competences.

Reviewer #2: The paper is very interesting and asks important questions. The paper is also well-motivated, explaining the context well and showing a good understanding of the Chinese context. It is actually quite valuable to have a study focusing on China. Here my specific comments:

The theoretical part was well executed. The hypotheses are interesting and well-developed. I only recommend relating things a little bit more to the literature on tax performance and tax effort. Moreover, it would be good to understand the study's relevance in the context of the overall literature that goes beyond China (that can also be done in the Conclusions). As for the competing hypotheses, I think it would be useful to attach the two oppositive hypotheses directly to the theoretical arguments instead of presenting the competing hypotheses at the end. Moreover, I was wondering, per se, whether there could be some sort of non-linearity.

The sample interval (2015-2019) due to Covid makes sense. Moreover, it is good to see that they construct two actual tax burden indicators.

Overall, the results are quite interesting. If you include macro variables, it might be useful to also check what happens if cluster over the macro unit (adjusted standard errors accordingly). It is also important that the paper discusses the quantitative effects (including "economic" significance). As for Table 3, I like how the variables have been added sequentially, which helps to check the robustness of the core govdebt1 variable.

Clarify a little bit better how you measure tax collection effort and tax incentive intensity. Similarly, there are different ways of measuring better institutional environments (see, e.g., governance indicators of the World Bank). Clarify how you measure and divide it. I struggle with the classification also when looking at the number of observations between both (five times higher for the better institutional environment, see Table 7). Maybe test the robustness with different cut-offs.

I don't like the wording "endogenetic test". Maybe better testing for endogeneity. It think, it would be useful to discuss the theoretical justification of the instrument a little bit more. Provide also some diagnostic tests.

It is good that some robustness tests were provided (see Table 9).

Conclusions. Good that some policy implications were provided. I recommend to add also a discussion of the study's limitations and future perspectives. Moreover, clarify again the innovative nature of the study and how the study fits into the overall literature.

Data always plural (see APA manual).

6. PLOS authors have the option to publish the peer review history of their article (what does this mean?). If published, this will include your full peer review and any attached files.

Reviewer #1: No

Reviewer #2: **Yes: **Benno Torgler

---

## [Author Response · Author response to Decision Letter 0]

25 Jan 2023

Submission Id: PONE-D-22-28623

Title: Local government debt and corporate tax burden: a perspective based on the trade-off of government tax collection and management

Dear Editors and Reviewers,

Thank you very much for your guidance and review of the article, and put forward very valuable review suggestions. These opinions have an important guiding role in the research design and theoretical analysis of the article, and greatly improve the article.

According to the suggestions of the reviewers, we have revised and replied each point which is being raised, and the revised part are marked in the file labeled 'Revised Manuscript with Track Changes'. The specific modification instructions are as follows:

For Reviewer #1:  Review of „Local government debt and corporate tax burden”

The paper looks at the relationship between the development of public debt on regional level in China and the efforts to collect more taxes from corporations. The hypothesis tested is whether the municipalities have incentives to increase their collection efforts as a result in increases in public debt to reduce the fiscal pressure. The authors find, using a panel regression with fixed effects and an instrumental variables specification, that higher levels of public debt correlate positively with the tax burden of companies.

While the topic is interesting and relevant, the methodology of investigation is weak and does not support the conclusions. Moreover, the results are very context specific, which raises doubts regarding the external validity and thus interest for a general-interest journal such as Plos One. I do not recommend publication of the paper.

Major suggestions: 

• Panel estimation: while the number of observations looks impressive, as the authors treat year-companies as a unit of observation, closer look reveals that the actual variation is much lower. Firstly, only five years are considered, which limits the time variation of the main explanatory variable. Secondly, and more importantly, local public debt is measures at the regional (local?) level (in fact, nowhere in the paper it is specified what the actual unit of observation of the debt variable is!). Implicitly, it seems that this is provincial data (e.g., referring to other provinces in the construction of the instrument), which means that the spatial variation is at 23 provinces or 34 regions. This does not allow for serious panel estimation. Moreover, this estimation requires an inclusion of regional fixed effects – which are not included. Why not?

Thanks to the reviewer for the suggestions. Limited by practical factors and data conditions, the research samples in this study are indeed not in a completely ideal research scenario. However, based on the reference and comparison of the existing literature, we believe that the empirical data in this study are based on the real and reliable realistic situation, and the research methods of the existing authoritative literature are used for reference as much as possible, which can provide empirical evidence for the research topic of this paper. The details are as follows:

(1)As China is in the stage of emerging and transition, the relevant systems of local government bond issuance mode are also in the process of continuous adjustment and improvement. Before 2015, China's central government did not strictly control the bond issuance mode of local governments, and local governments had a variety of hidden financing channels, resulting in multiple statistical caliber of local government debt data, and the data were not open and transparent to the public. Since 2015, China's new Securities Law has been implemented, which strictly requires local governments to borrow debt by the “self-issue and self-repayment” model. And the relevant data are disclosed on China Government Debt Center (http://www.celma.org.cn/), which is more authoritative and reliable than the estimated and manually collected data in previous literature. Therefore, we have to define the research interval from 2015 to 2019. Although the research interval is short, a total of 12,325 company-year observations are provided by 31 provinces and regions in the five years, which can provide sufficient empirical evidence for the research of this paper.

(2)This paper explains the scope of the sample in the sample selection section of the research design. Specifically, the samples come from the debt balance information of all provinces and cities in China since 2015, including 23 provinces, 4 municipalities directly under the Central Government and 5 autonomous regions. Combining the data of local government debt at the macro level with the data of enterprises at the micro level, this paper examines the microeconomic consequences of local government debt by referring to many authoritative literature, such as Croce et al. (2019), Liang et al. (2017) and Rao et al. (2022), which can provide empirical evidence for the study. 

The main reasons for not including regional fixed effects are as follows:

In China, the statutory corporate tax rate and tax categories are uniformly set by the central government, and there are no significant differences between different regions of the country. Changes in the actual tax burden level will be around the statutory tax rate level, and the degree of fluctuation is small, so it is difficult to observe changes in the tax burden level of enterprises in the same region in each year. In addition, this paper intended to use the differences between different regions to observe the tax trade-off of different local governments. The setting of the model can help us to start from the overall perspective and observe that when the statutory tax rate is basically the same, the regions with larger local government debt have higher actual tax burden. Based on the above considerations, this paper does not include regional fixed effects. However, we controlled macro-level factors as much as possible, such as regional population, investment status and regional economic status. Moreover, the standard errors are clustered to the provincial and municipal levels, which can control the related interference factors to a certain extent.

[1] Liang Y, Shi K, Wang L, Xu J. Local Government Debt and Firm Leverage: Evidence from China. Asian Econ Policy Rev. 2017; 12(2):210-232. https://10.1111/aepr.12176.

[2] Croce MM, Nguyen TT, Raymond S, Schmid L. Government debt and the returns to innovation. J Financ Econ. 2019; 132(3):205-225. https://10.1016/j.jfineco.2018.11.010.

[3] Rao PG, Tang S, Li XX. The Crowding-Out Effect of Local Government Debt:Evidence from Corporate Leverage Manipulation. China Industrial Economics. 2022;(01):151-169. https://10.19581/j.cnki.ciejournal.2022.01.009.

• Related to the previous point is the question of endogeneity. An obvious candidate for a confounder is the overall economic activity in a region, which affects both the denominator of the debt ratio and (indirectly) the denominator of the tax burden, e.g., through operating income or pre-tax profit. This means, that any changes to the economic situation of each province, will likely affect both sides of the regression equation. Authors use in one specification a Hausman-type spatial instrument, which replaces own debt levels, with an average of debt levels of other provinces. This instrument does not resolve the endogeneity issue, as any idiosyncratic change to the economic situation of all regions still affects both the explanatory and the outcome variables. Besides, no test statistics of the instruments, such as first-stage regressions, or the overidentification tests, F-Statistics and like to assess the quality of the instrument are provided.

Thanks to the reviewer for the suggestions. 

Drawing on the existing research (Demirci et al., 2019; Chongi et al., 2013), we adjusted the instrumental variables for the articles:

we use the mean value of education expenditure of other provinces and cities in that year as the instrumental variable (Espend). On the one hand, the evaluation system of local officials in China includes the inspection of people's livelihood matters, so there is competition among local governments in the scale of fiscal expenditure on people's livelihood (Lu et al., 2022). That is, when other regions increase fiscal expenditure to improve people's livelihood performance, local governments will increase fiscal expenditure accordingly for regional competition. Moreover, there is a certain correlation between government revenue and expenditure and local government debt scale, which meets the correlation requirements of instrumental variables. On the other hand, education expenditure is the basic expenditure of local finance on people's livelihood, which belongs to the local rigid expenditure and has a weak correlation with the short-term economic fluctuations that affect the actual tax burden of enterprises. This means Espend is relatively exogenous. At the same time, the pressure of fiscal expenditure in other regions will not be directly transferred to enterprises within the jurisdiction. All the above characteristics enable Espend to meet the exogeneity requirements of instrumental variables.

Table 6 reports the regression results of the instrumental variables in Column (1)-(4). And the instrumental variable (Espend) passed the underidentification test and the weak identification test. 

Table 6. Testing for endogeneity.

 (1) (2) (3) (4)

 first second first second

Variables GovDebt1 Taxburden1 GovDebt1 Taxburden2

Espend 0.002*** 0.002*** 

 (7.47) (7.67) 

GovDebt1 0.0766*** 0.0625**

 (3.98) (2.98)

Namerate 0.029 0.559*** 0.003 0.117***

 (0.67) (13.68) (0.10) (6.14)

Size 0.000 -0.002 -0.001 0.002**

 (-0.09) (-1.51) (-0.68) (2.72)

Roa -0.053 -0.608*** -0.034 0.268***

 (-1.41) (-11.63) (-0.97) (11.97)

BM -0.004 0.014 0.007 -0.002 

 (-0.60) (1.91) (0.97) (-0.49)

Lev 0.010 0.011 0.010 -0.051***

 (1.13) (1.32) (1.17) (-11.61)

Tangible 0.006 0.002 0.008 -0.008 

 (0.54) (0.18) (0.68) (-1.38)

Intangible -0.023 0.056 -0.020 0.057**

 (-0.73) (1.71) (-0.71) (3.08)

Invent -0.021 0.034** -0.022 -0.011 

 (-1.20) (2.71) (-1.34) (-1.20)

Loss 0.004 0.036*** 0.005 -0.004*

 (1.49) (4.43) (1.72) (-2.11)

Top10 -0.014 -0.003 -0.009 0.010*

 (-1.75) (-0.39) (-0.91) (2.25)

Indep 0.025 0.049** 0.016 0.006 

 (0.79) (2.74) (0.68) (0.47)

Density -0.273*** -0.002 0.000*** 0.000 

 (-3.66) (-0.30) (-3.74) (-1.88)

Inv 0.208 -0.058 0.132 -0.130***

 (0.58) (-0.83) (0.38) (-3.61)

GDPr -0.345** 0.014 -0.354** 0.018 

 (-3.05) (0.67) (-3.12) (1.35)

Constant -1.531*** 0.041 -1.495*** -0.037*

 (-6.27) (1.13) (-6.69) (-1.99)

Year/Industry YES YES YES YES

Observations 12,325 12,325 9,390 9,390

R2 0.263 0.302 

Adj-R2 0.536 0.564 

Underidentification test: 

Kleibergen-Paap rk LM statistic 5.055** 5.275**

Weak identification test: 

Kleibergen-Paap Wald rk F statistic 55.84*** 58.84***

Overidentification test of all instruments: 

Hansen J statistic equation exactly identified equation exactly identified

Notes: *, ** and *** indicate statistical significance at the 10, 5, and 1% levels, respectively. The values in parentheses in columns (1) and (3) are t-statistics, and the values in parentheses in columns (2) and (4) are z-statistics. The standard errors are clustered at the provincial and municipal levels (31 groups) by using the wild cluster bootstrap method.

[1] Demirci I, Huang J, Sialm C. Government debt and corporate leverage: International evidence. J Financ Econ. 2019; 133(2):337-356. https://10.1016/j.jfineco.2019.03.009.

[2] Chong TT, Lu L, Ongena S. Does banking competition alleviate or worsen credit constraints faced by small- and medium-sized enterprises? Evidence from China. J Bank Financ. 2013; 37(9):3412-3424. https://10.1016/j.jbankfin.2013.05.006.

[3] Lu J, Cai S, Zang T. Strategic Competition of County-level Governments under Promotion Game:Evidence from Zhejiang Province. Finance and Trade Research. 2022; 33(05):1-14. https://10.19337/j.cnki.34-1093/f.2022.05.001.

• The overall empirical exercise is very context specific. The authors mention, that the channel of transmission for the higher tax burden is essentially only the tax collection effort, as the taxation rates and bases as such as regulated nation-wide. This raises the question of external validity – what can we learn from the paper on the similar cases all over the world?

Thanks to the reviewer for the suggestions. 

Indeed, this paper takes the realistic situation in China as the specific research background, but the phenomena involved in this paper, such as government intervention, taxation trade-off, and regional competition, exist widely in all countries in the world (including developed and developing countries). With the help of the realistic situation of China, a representative developing country, this article enriches and expands the literature in the above fields, which has important theoretical and practical significance. The details are as follows:

Taking the actual tax burden level of micro enterprises as the research object, this paper expands the research on the economic consequences of local governments' expanding bond issuance in developing countries. At the same time, through mechanism test, heterogeneity analysis and other further tests, this paper reveals the role of local governments' intervention behavior in this process, analyzes the trade-off of local governments' taxation under different institutional environments, and enriches the literature in tax collection, regional competition and other related fields. It provides policy enlightenment for developing countries with large differences in regional development levels to improve public debt management system, create a fair tax environment, and promote high-quality economic growth, which has important theoretical and practical significance. 

In addition, we have added a description of the above research contributions at the end of the article.

• Hypotheses 2 and 3 have left me a bit puzzled. The authors suggest that the impact of debt on tax burden should be *more significant* in non-state-owned enterprises. I would think that there should *only* be an effect in those companies, unless there is something about China, which makes is “normal” that state-owned entreprises do not fulfill their tax obligations. If this is indeed the case, it needs to be explained in more detail.

Thanks to the reviewer for the suggestions. 

As mentioned in the article, there is a natural blood relationship between state-owned enterprises and the government in China. Many studies have shown that (Chen et al., 2016; Liu and Hou, 2017; Yan et al., 2021), on the one hand, state-owned enterprises undertake a large number of political tasks such as public services, national defense construction, people's livelihood improvement, and stable employment. Therefore, the government will provide more policy compensation to them, including providing more tax incentives and a more relaxed tax collection environment. On the other hand, Chinese state-owned enterprises have strong tax lobbying capabilities. In order to ensure the financial flexibility and daily operation of enterprises, they will seek support from local governments to reduce the tax burden when necessary. And local governments are willing to provide a more lenient tax environment to state-owned enterprises in order to protect them and ensure their continued operation. Therefore, there is also room for adjustment of the tax burden of state-owned enterprises in China.

[1] Chen D, Kong MQ, Wang HJ. Economic cycle and tax avoidance of state-owned enterprises. Management World. 2016;(05):46-63. https://10.19744/j.cnki.11-1235/f.2016.05.006.

[2] Liu SY, Hou DC. The Measurement of Tax Bargaining Power of Listed Companies and Government——An Empirical Study Based on Bilateral Stochastic Frontier Model. Taxation and Economy. 2017;(04):87-95.

 [3] Yan JM, Liu JJ, Yan H. Tax Burden and Tax Preferences from the Perspective of Tax Neutrality in China. Taxation and Economy. 2021;(3):22-31.

Minor suggestions: 

• Page 3, line 38: what are financing platforms?

Local government financing platforms—companies capitalized and owned by local government and established for the purpose of raising funds for municipal infrastructure construction—emerged in China in the 1980s as a response to the severe constraints on indebtedness by local governments themselves.

These local government off-balance-sheet companies circumvented the 1994 Budget Law regulation prohibiting local governments from market borrowing and raised funds through various means, including bank loans, urban investment bonds, and the so-called yinxinzheng, that is, investment from investment trust companies (Cheng et al., 2022). Before the implementation of the new Budget Law in 2015, local governments mainly used financing platforms to raise funds, which was the main driving force for the expansion of local government debt in China. Since the relevant bond issuance data of local financing platforms have not been disclosed, and the central government has standardized the issuance mode of local bonds after 2015, that is, local governments are strictly prohibited from borrowing debt through financing platforms. Therefore, this paper does not give a detailed description of such debt, and only describes the existence of the platform in part of the description of the reality. For information on local financing platforms, please refer to Tao (2015), Clarke and Lu (2017) and Cai et al. (2021).

[1]Cheng, Y. , Jia, S. , & Meng, H. Fiscal policy choices of local governments in China: Land finance or local government debt? [J]. International Review of Economics & Finance, 2022, 80: 294-308.

[2]Tao, K. Assessing Local Government Debt Risks in China: A Case Study of Local Government Financial Vehicles [J]. China & World Economy, 2015, 23(5): 1-25.

[3]Clarke, D. , & Lu, F. The Law of China’s Local Government Debt: Local Government Financing Vehicles and Their Bonds [J]. The American Journal of Comparative Law, 2017, 65(4): 751-798.

[4]Cai, M. , Fan, J. , Ye, C. , & Zhang, Q. Government debt, land financing and distributive justice in China [J]. Urban Studies, 2021, 58(11): 2329-2347.

• Pge 8, line 155: what is land finance? Does this refer to management of municipal property?

Land assets are an important source of local government finance in most developing countries (Cheng et al., 2022). China's land finance refers to that local governments obtain high land transfer revenue through the transfer of land-use rights, which provides a large amount of financial support for local development and is an important part of local governments' fiscal revenue. At the same time, local governments mortgage land to banks to obtain land mortgage loans and rely on land to borrow to repay debts, forming a "land-infrastructure-leverage" strategy in urban development, resulting in the continuous expansion of local government debt.

The purpose of mentioning the term "land finance" in this paper is only to state the basic situation of local finance in China, which does not affect the main content of this paper, so it is not described in detail.

[1] Cheng, Y. , Jia, S. , & Meng, H. Fiscal policy choices of local governments in China: Land finance or local government debt? [J]. International Review of Economics & Finance, 2022, 80: 294-308.

• I don’t understand the difference between the local and state taxation bureaus and their competences.

Thanks to the reviewer for the suggestions. 

The narrative about the difference between Local Taxation Bureas and State Taxation Bureas has been supplemented in the newly revised manuscript. The details are as follows:

Since the reform of the tax distribution system in 1994, China has implemented a parallel system of the State Taxation Bureau (STB) and the Local Taxation Bureau (LTB) for a long time. The STB is directly under the State Administration of Taxation (which is under the State Council of the People's Republic of China), while the LTB is under the dual leadership of the local government and the State Administration of Taxation, and the main administrative authority is assigned to the local government. Therefore, the strength of tax collection and administration of the STB and the LTB reflects the will of different leading departments and there are significant differences. To be specific, although the departments of STB in some regions jointly work with local financial departments and rely on local governments to provide basic office conditions (Sun, 2017), the STB are directly under the State Administration of Taxation and are subject to limited intervention by local governments, while the organizational structure, staffing and management system of the LTB are all responsible for by the local government, so they are often subject to greater intervention by local governments (Yang et al.,2020).

[1] Sun G. Tax Enforcement and Capital Investment Efficiency of Chinese Listed Firms:Preliminary Evidences of Irregular Taxes Rebate or Reimbursement from Local Governments. Journal of Central University of Finance & Economics. 2017;(11):3-17.

[2] Yang XD, Shen YJ, Peng CC. Will Environmental Investment Affects the Firm's Actual Tax Burden?——Evidence from Heavily Polluting Industries. Accounting Research. 2020;(05):134-146. https://10.3969/j.issn.1003-2886.2020.05.010.

For Reviewer #2:  

Thank you very much for your guidance and review of the article, and put forward very valuable review suggestions. These opinions have an important guiding role in the research design and theoretical analysis of the article, and greatly improve the article.

According to the suggestions, we have revised and replied each point which is being raised, and the revised part are marked in the file labeled 'Revised Manuscript with Track Changes'. The specific modification instructions are as follows:

The paper is very interesting and asks important questions. The paper is also well-motivated, explaining the context well and showing a good understanding of the Chinese context. It is actually quite valuable to have a study focusing on China. Here my specific suggestions: 

• The theoretical part was well executed. The hypotheses are interesting and well-developed. I only recommend relating things a little bit more to the literature on tax performance and tax effort. Moreover, it would be good to understand the study's relevance in the context of the overall literature that goes beyond China (that can also be done in the Conclusions). As for the competing hypotheses, I think it would be useful to attach the two oppositive hypotheses directly to the theoretical arguments instead of presenting the competing hypotheses at the end. Moreover, I was wondering, per se, whether there could be some sort of non-linearity.

Thank you very much for your recognition and suggestions. We have revised the article according to the suggestions. To be specific:

(1)In the theoretical analysis part, we reorganize the research ideas. Then, in the context of the overall literature, we add relevant literature review and theoretical derivation of local governments' intervention in tax collection when facing fiscal pressure. At the same time, we also emphasize this in the conclusion section at the end of the paper.

(2) The position of the competing hypotheses is adjusted. 

(3) In addition, we are very grateful to the reviewer for the new idea. We try to observe the nonlinear relationship between variables, but the empirical results show that there is no nonlinear relationship in this paper.

• The sample interval (2015-2019) due to Covid makes sense. Moreover, it is good to see that they construct two actual tax burden indicators.

• Overall, the results are quite interesting. If you include macro variables, it might be useful to also check what happens if cluster over the macro unit (adjusted standard errors accordingly). It is also important that the paper discusses the quantitative effects (including "economic" significance). As for Table 3, I like how the variables have been added sequentially, which helps to check the robustness of the core govdebt1 variable.

Thank you very much for your recognition and suggestions. We have revised the article according to the suggestions. To be specific:

(1)The standard errors of all tests in this paper are clustered to the macro provincial and municipal levels, and the detailed regression results are shown in the manuscript. 

(2)We add the economic significance analysis in the main test section.

• Clarify a little bit better how you measure tax collection effort and tax incentive intensity. Similarly, there are different ways of measuring better institutional environments (see, e.g., governance indicators of the World Bank). Clarify how you measure and divide it. I struggle with the classification also when looking at the number of observations between both (five times higher for the better institutional environment, see Table 7). Maybe test the robustness with different cut-offs.

Thanks to the reviewer for the suggestions. We have revised the article according to the suggestions. To be specific:

(1) We supplement the measurement description of the intensity of tax collection effort and tax incentive intensity.

(2) For grouping by institutional environment, we use the index reports used in the vast majority of the Chinese regional studies literature (Wang et al., 2019). Established in 2000, the index system systematically measures the marketization progress, market development degree and legal environment of 31 provinces and cities in China from different aspects, which is a systematic and complete index system.

Initially, we were also surprised by the number of groupings of institutional environments. The quantitative gap between the two is indeed very large. Therefore, we carefully observed and explored the reasons for the large gap in the number of groups. We find that, because the regions with poor institutional environment cannot provide a better business environment for enterprises and lack economic attractiveness, the number of listed companies in these regions is small. And this phenomenon also proves the research logic of this paper. It is precisely because of the small number of companies in such areas that local governments are more worried about the outflow of the liquid tax base and are more willing to provide a relaxed tax collection environment to retain enterprises.

In order to ensure the robustness of the results, in the new revised manuscript, we report the regression results grouped by the tertile of the indicator. After the regrouping, the gap in the number of sample observations is narrowed, and the original conclusion is still supported.

[1] Wang XL, Fan G, Hu LP. (2019). MARKETIZATON INDEX OF CHINA'S PROVINCES:NERI REPORT 2018. Beijing: Economic Science Press

• I don't like the wording "endogenetic test". Maybe better testing for endogeneity. It think, it would be useful to discuss the theoretical justification of the instrument a little bit more. Provide also some diagnostic tests.

Thanks to the reviewer for the suggestions. We replaced the instrumental variables in the manuscript and reported the relevant diagnostic tests. To be specific: 

Drawing on the existing research (Demirci et al., 2019; Chongi et al., 2013), we adjusted the instrumental variables for the articles:

we use the mean value of education expenditure of other provinces and cities in that year as the instrumental variable (Espend). On the one hand, the evaluation system of local officials in China includes the inspection of people's livelihood matters, so there is competition among local governments in the scale of fiscal expenditure on people's livelihood (Lu et al., 2022). That is, when other regions increase fiscal expenditure to improve people's livelihood performance, local governments will increase fiscal expenditure accordingly for regional competition. Moreover, there is a certain correlation between government revenue and expenditure and local government debt scale, which meets the correlation requirements of instrumental variables. On the other hand, education expenditure is the basic expenditure of local finance on people's livelihood, which belongs to the local rigid expenditure and has a weak correlation with the short-term economic fluctuations that affect the actual tax burden of enterprises. This means Espend is relatively exogenous. At the same time, the pressure of fiscal expenditure in other regions will not be directly transferred to enterprises within the jurisdiction. All the above characteristics enable Espend to meet the exogeneity requirements of instrumental variables.

Table 6 reports the regression results of the instrumental variables in Column (1)-(4). And the instrumental variable (Espend) passed the underidentification test and the weak identification test. 

Table 6. Testing for endogeneity.

 (1) (2) (3) (4)

 first second first second

Variables GovDebt1 Taxburden1 GovDebt1 Taxburden2

Espend 0.002*** 0.002*** 

 (7.47) (7.67) 

GovDebt1 0.0766*** 0.0625**

 (3.98) (2.98)

Namerate 0.029 0.559*** 0.003 0.117***

 (0.67) (13.68) (0.10) (6.14)

Size 0.000 -0.002 -0.001 0.002**

 (-0.09) (-1.51) (-0.68) (2.72)

Roa -0.053 -0.608*** -0.034 0.268***

 (-1.41) (-11.63) (-0.97) (11.97)

BM -0.004 0.014 0.007 -0.002 

 (-0.60) (1.91) (0.97) (-0.49)

Lev 0.010 0.011 0.010 -0.051***

 (1.13) (1.32) (1.17) (-11.61)

Tangible 0.006 0.002 0.008 -0.008 

 (0.54) (0.18) (0.68) (-1.38)

Intangible -0.023 0.056 -0.020 0.057**

 (-0.73) (1.71) (-0.71) (3.08)

Invent -0.021 0.034** -0.022 -0.011 

 (-1.20) (2.71) (-1.34) (-1.20)

Loss 0.004 0.036*** 0.005 -0.004*

 (1.49) (4.43) (1.72) (-2.11)

Top10 -0.014 -0.003 -0.009 0.010*

 (-1.75) (-0.39) (-0.91) (2.25)

Indep 0.025 0.049** 0.016 0.006 

 (0.79) (2.74) (0.68) (0.47)

Density -0.273*** -0.002 0.000*** 0.000 

 (-3.66) (-0.30) (-3.74) (-1.88)

Inv 0.208 -0.058 0.132 -0.130***

 (0.58) (-0.83) (0.38) (-3.61)

GDPr -0.345** 0.014 -0.354** 0.018 

 (-3.05) (0.67) (-3.12) (1.35)

Constant -1.531*** 0.041 -1.495*** -0.037*

 (-6.27) (1.13) (-6.69) (-1.99)

Year/Industry YES YES YES YES

Observations 12,325 12,325 9,390 9,390

R2 0.263 0.302 

Adj-R2 0.536 0.564 

Underidentification test: 

Kleibergen-Paap rk LM statistic 5.055** 5.275**

Weak identification test: 

Kleibergen-Paap Wald rk F statistic 55.84*** 58.84***

Overidentification test of all instruments: 

Hansen J statistic equation exactly identified equation exactly identified

Notes: *, ** and *** indicate statistical significance at the 10, 5, and 1% levels, respectively. The values in parentheses in columns (1) and (3) are t-statistics, and the values in parentheses in columns (2) and (4) are z-statistics. The standard errors are clustered at the provincial and municipal levels (31 groups) by using the wild cluster bootstrap method.

[1] Demirci I, Huang J, Sialm C. Government debt and corporate leverage: International evidence. J Financ Econ. 2019; 133(2):337-356. https://10.1016/j.jfineco.2019.03.009.

[2] Chong TT, Lu L, Ongena S. Does banking competition alleviate or worsen credit constraints faced by small- and medium-sized enterprises? Evidence from China. J Bank Financ. 2013; 37(9):3412-3424. https://10.1016/j.jbankfin.2013.05.006.

[3] Lu J, Cai S, Zang T. Strategic Competition of County-level Governments under Promotion Game:Evidence from Zhejiang Province. Finance and Trade Research. 2022; 33(05):1-14. https://10.19337/j.cnki.34-1093/f.2022.05.001.

• It is good that some robustness tests were provided (see Table 9).

• Conclusions. Good that some policy implications were provided. I recommend to add also a discussion of the study's limitations and future perspectives. Moreover, clarify again the innovative nature of the study and how the study fits into the overall literature.

Thanks to the reviewer for the recognition and suggestions. 

In the conclusion section, we sort out and clarify the research contribution of this paper against the background of the overall literature, and add the discussion of research limitations and future prospects. The details are as follows:

Taking the actual tax burden level of micro enterprises as the research object, this paper expands the research on the economic consequences of local governments' expanding bond issuance in developing countries. At the same time, through mechanism test, heterogeneity analysis and other further tests, this paper reveals the role of local governments' intervention behavior in this process, analyzes the trade-off of local governments' taxation under different institutional environments, and enriches the literature in tax collection, regional competition and other related fields. It provides policy enlightenment for developing countries with large differences in regional development levels to improve public debt management system, create a fair tax environment, and promote high-quality economic growth, which has important theoretical and practical significance. However, due to the limitations of data and research conditions, there are still some problems that need to be further explored. First of all, due to the lack of data, the regional tax collection intensity index in this paper refers to the estimation method of the existing research, which lacks the examination of the changes in the actual tax collection behavior of the tax authorities. If we can directly describe the intensity of tax collection through the tax authorities' behaviors (such as strengthening pre-supervision, in-process investigation and post-inspection), we can more accurately and carefully observe the impact of local government debt expansion on regional tax collection and administration, and put forward corresponding policy suggestions. Second, the research conclusion of this paper shows that local governments will intervene in tax collection when faced with expanding debt scale, which will increase the tax burden of enterprises within their jurisdiction. However, due to the lack of data, the analysis has not been carried out from other perspectives, such as whether local governments support industries with higher tax contribution in order to obtain more tax revenue, and how underdeveloped regions that do not obtain fiscal revenue growth by intervening in tax collection deal with the pressure of debt expansion in the short term. The above issues still need further discussion in the future.

• Data always plural (see APA manual).

Thanks for your careful review, we have corrected this issue.

---

## [Decision Letter · Decision Letter 1]

4 Apr 2023

PONE-D-22-28623R1Local government debt and corporate tax burden: a perspective based on the trade-off of government tax collection and managementPLOS ONE

Dear Dr. Zhao,

Thank you for submitting your manuscript to PLOS ONE. After careful consideration, we feel that it has merit but does not fully meet PLOS ONE’s publication criteria as it currently stands. Therefore, we invite you to submit a revised version of the manuscript that addresses the points raised during the review process.

Thank you very much for revising your manuscript. I see that the paper has improved significantly. However, there are still some points open that I would like to see addressed in a further revision. Reviewer 1 is still a bit skeptical and recommends further robustness checks. I support this. The reviewer makes concrete suggestions here, which I consider useful. Reviewer 2 recommends a proof of the paper by a native speaker, which I also expect. Like reviewer 2, I would also be pleased if the data were made publicly available to the community.

We look forward to receiving your revised manuscript.

Kind regards,

Florian Follert

Academic Editor

PLOS ONE

Journal Requirements:

Reviewers' comments:

Reviewer's Responses to Questions

**Comments to the Author**

1. If the authors have adequately addressed your comments raised in a previous round of review and you feel that this manuscript is now acceptable for publication, you may indicate that here to bypass the “Comments to the Author” section, enter your conflict of interest statement in the “Confidential to Editor” section, and submit your "Accept" recommendation.

Reviewer #1: All comments have been addressed

Reviewer #2: All comments have been addressed

2. Is the manuscript technically sound, and do the data support the conclusions?

Reviewer #1: Partly

Reviewer #2: Yes

3. Has the statistical analysis been performed appropriately and rigorously? 

Reviewer #1: Yes

Reviewer #2: Yes

4. Have the authors made all data underlying the findings in their manuscript fully available?

Reviewer #1: Yes

Reviewer #2: No

5. Is the manuscript presented in an intelligible fashion and written in standard English?

Reviewer #1: Yes

Reviewer #2: No

6. Review Comments to the Author

Reviewer #1: see the attached file

Reviewer #2: I am happy with the adjustments based on my comments provided. I recommend to make the data and the codes available (e.g., via OSF, https://osf.io/). I also recommend to proofread the article by native English speaker.

7. PLOS authors have the option to publish the peer review history of their article (what does this mean?). If published, this will include your full peer review and any attached files.

Reviewer #1: No

Reviewer #2: No

---

## [Author Response · Author response to Decision Letter 1]

8 Jun 2023

Response to Reviewers

Submission Id: PONE-D-22-28623

Title: Local government debt and corporate tax burden: a perspective based on the trade-off of government tax collection and management

Dear Editors and Reviewers,

Thank you very much for your guidance and review of the article, and put forward very valuable review suggestions. According to the suggestions of the reviewers, we have revised or replied each point which is being raised, and the revised part are marked in the file labeled 'Revised Manuscript with Track Changes'. The specific modification instructions are as follows:

For Reviewer #1:  

• Thank you for clarifying the limited sample size. Time restriction now makes more sense to me. While I am still not entirely convinced by the fact that there is very little variation, it is good to see that the coefficients remain significant when clustering at the provincial level. However, as authors mention, there is little to no variation of the main independent variable over time (which is why they do not include fixed effects), the empirical exercise is essentially a cross-unit regression on 32 units of observations. Such a set-up could be very sensitive to outliers. You should provide some tests, e.g., residuals vs leverage plots or (better) as an additional robustness “leave-one-out” regressions.

Thanks to the reviewer for the suggestions. We tested for the presence of outliers in the main regression by constructing Residuals vs Leverage Plots. The results are shown in Figure 1 (dependent variable: Taxburden1) and Figure 2 (dependent variable: Taxburden2). As can be seen from the plot, there are no leverage points exceeding 0.5 in our sample (meeting the threshold requirement for Cook's Distance), indicating once again that our results are relatively robust.

Fig1

Fig2

• I like the idea of the education expenditure instrument and it is good that you now provided some diagnostic tests. I do not particularly like the wording “testing for endogeneity” – you cannot test for endogeneity – but the contents of the table are solid.

We appreciate the recognition from the reviewer regarding our work. The wording "testing for endogeneity" comes from another expert's suggestion in the previous round of review. This time, we have collectively included the issue of endogeneity under robustness testing and revised the section addressing endogeneity as "Endogeneity discussion."

• It is now more clear how the research adds to the literature.

We greatly appreciate the recognition of our work by the reviewer.

• This is now more clear

We greatly appreciate the recognition of our work by the reviewer.

Minor suggestions: 

• Table 6 is beyond the margins and the righthand side is, thus, missing.

Thank you very much for your thorough review. We have made adjustments to the display of the table.

For Reviewer #2:  

Thank you very much for your guidance and review of the article, and put forward very valuable review suggestions. According to the suggestions of the reviewers, we have revised or replied each point which is being raised, and the revised part are marked in the file labeled 'Revised Manuscript with Track Changes'. The specific modification instructions are as follows:

• I am happy with the adjustments based on my comments provided. I recommend to make the data and the codes available (e.g., via OSF, https://osf.io/). I also recommend to proofread the article by native English speaker.

We appreciate the recognition of our work by the reviewer.

Regarding the data section of the manuscript, we have disclosed all data sources in the "Data Availability Statement" section. This study analyzed publicly available datasets. The financial data of companies was obtained from the CSMAR database (https://www.gtarsc.com/); data related to local government debt was sourced from the China Government Debt Center (http://www.celma.org.cn/); other macroeconomic data was obtained from the National Bureau of Statistics of China (https://data.stats.gov.cn/).

It should be noted that when using the CSMAR database, it is emphasized that "no unit or individual may modify, use, copy, intercept, compile, edit, upload, download, or in any way or medium reproduce, reprint, or disseminate any part of this software work without the written authorization of Shenzhen Securities Information Co., Ltd. for any purpose (including but not limited to commercial use and non-commercial purposes such as learning and research). Otherwise, it will be deemed as an infringement, and Shenzhen Securities Information Co., Ltd. reserves the right to pursue legal responsibilities in accordance with the law."

Therefore, we do not have the right to forward or disclose this portion of the data, but we have provided the detailed address of the database for downloading purposes.

Regarding the language of the article, we have requested assistance from a native English speaker to review and revise the article, and we have asked her to provide a proof.

For Editors:  

Thank you for submitting your manuscript to PLOS ONE. After careful consideration, we feel that it has merit but does not fully meet PLOS ONE’s publication criteria as it currently stands. Therefore, we invite you to submit a revised version of the manuscript that addresses the points raised during the review process.

Thank you very much for revising your manuscript. I see that the paper has improved significantly. However, there are still some points open that I would like to see addressed in a further revision. Reviewer 1 is still a bit skeptical and recommends further robustness checks. I support this. The reviewer makes concrete suggestions here, which I consider useful. Reviewer 2 recommends a proof of the paper by a native speaker, which I also expect. Like reviewer 2, I would also be pleased if the data were made publicly available to the community.

Thank you for your suggestions. We have reviewed all the references in the article, and none of them have been retracted. However, some of the references do not have DOI addresses. These references play an important role in providing evidence and logical analysis for the article, so we have not removed them. Here are the URLs of such references.

1.Zhu J, Kou F. The Effects of Local Government Debt on Total Factor Productivity in China ——On the Driving Force of Local Government Debt Expansion: The Strong or the Weak. Journal of Hebei University (Philosophy and Social Science). 2019; 44(06):80-92. 

https://xb-zsb.hbu.edu.cn/CN/10.3969/j.issn.1005-6378.2019.06.010

14.Sun G. Tax Enforcement and Capital Investment Efficiency of Chinese Listed Firms:Preliminary Evidences of Irregular Taxes Rebate or Reimbursement from Local Governments. Journal of Central University of Finance & Economics. 2017;(11):3-17

https://kns.cnki.net/kcms2/article/abstract?v=3uoqIhG8C44YLTlOAiTRKibYlV5Vjs7iAEhECQAQ9aTiC5BjCgn0Rk8WheNbdLji-d5UwnULCUtOs9pWN41Vy30hVBAh85CL&uniplatform=NZKPT

43. Xu J, Wang X, He Z. Multidimensional Performance Appraisals, Chinese Government Competition and Local Tax Administration. Economic Research Journal. 2019; 54(04):33-48.

https://kns.cnki.net/kcms2/article/abstract?v=3uoqIhG8C44YLTlOAiTRKibYlV5Vjs7iLik5jEcCI09uHa3oBxtWoHGwy9XepWzLhDkBIAV547T89QDb0bKgssPg7TXmX1iB&uniplatform=NZKPT

51.Mertens JB. Measuring tax effort in Central and Eastern Europe. Public finance and management. 2003; 3(4):530-563.

https://www.academia.edu/download/50939471/Measuring_Tax_Effort.pdf

---

## [Editor Report · Decision Letter 2]

13 Jun 2023

Local government debt and corporate tax burden: A perspective based on the trade-off of government tax collection and management

PONE-D-22-28623R2

Dear Dr. Zhao,

We’re pleased to inform you that your manuscript has been judged scientifically suitable for publication and will be formally accepted for publication once it meets all outstanding technical requirements.

Kind regards,

Florian Follert

Academic Editor

PLOS ONE

---

## [Editor Report · Acceptance letter]

4 Jul 2023

PONE-D-22-28623R2 

Local government debt and corporate tax burden: A perspective based on the trade-off of government tax collection and management 

Dear Dr. Zhao:

I'm pleased to inform you that your manuscript has been deemed suitable for publication in PLOS ONE. Congratulations! Your manuscript is now with our production department. 

Kind regards, 

on behalf of

Prof. Dr. Florian Follert 

Academic Editor

PLOS ONE